



**Modelling northern peatlands area and carbon dynamics since the Holocene with**
**the ORCHIDEE-PEAT land surface model (SVN r5488)**
Chunjing Qiu[1], Dan Zhu[1], Philippe Ciais[1], Bertrand Guenet[1], Shushi Peng[2], Gerhard
Krinner[3], Ardalan Tootchi[4], Agnès Ducharne[4], Adam Hastie[5],
1. Laboratoire des Sciences du Climat et de l'Environnement, UMR8212, CEA-CNRS-UVSQ F-
91191 Gif sur Yvette, France
2. Sino-French Institute for Earth System Science, College of Urban and Environmental Sciences,
Peking University, 100871 Beijing, China
3. CNRS, Université Grenoble Alpes, Institut de Géosciences de l'Environnement (IGE), F-38000
Grenoble, France
4. Sorbonne Université, CNRS, EPHE, Milieux environnementaux, transferts et interaction dans
les hydrosystèmes et les sols, Metis, F-75005 Paris, France
5. Department of Geoscience, Environment and Society, Université Libre de Bruxelles, 1050
Bruxelles, Belgium
Correspondence: Chunjing Qiu (chunjing.qiu@lsce.ipsl.fr)
**Abstract**
The importance of northern peatlands in the global carbon cycle has recently been
recognized, especially for long-term changes. Yet, the complex interactions between
climate and peatland hydrology, carbon storage and area dynamics make it challenging
to represent these systems in land surface models. This study describes how peatland
are included as an independent sub-grid hydrological soil unit (HSU) into the
ORCHIDEE-MICT land surface model. The peatland soil column in this tile is
characterized by multi-layered vertical water and carbon transport, and peat-specific
hydrological properties. A cost-efficient TOPMODEL approach is implemented to
simulate the dynamics of peatland area, calibrated by present-day wetland areas that are
regularly inundated or subject to shallow water tables. The model is tested across a
range of northern peatland sites and for gridded simulations over the Northern
Hemisphere (>30 °N). Simulated northern peatland area (3.9 million km$^2$), peat carbon
stock (463 PgC) and peat depth are generally consistent with observed estimates of
peatland area (3.4 – 4.0 million km$^2$), peat carbon (270 – 540 PgC) and data
compilations of peat core depths. Our results show that both net primary production

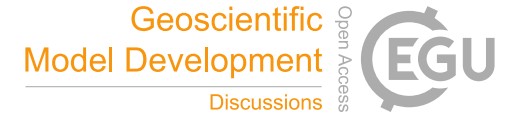

(NPP) and heterotrophic respiration (HR) of northern peatlands increased over the past
century in response to $CO_2$ and climate change. NPP increased more rapidly than HR,
and thus net ecosystem production (NEP) exhibited a positive trend, contributing a
cumulative carbon storage of 11.13 Pg C since 1901, most of it being realized after the
1950s.

## 1. Introduction

Northern peatlands carbon (C) stock is estimated between 270 and 540 PgC across an
area of 3.4 – 4 million $km^2$ (Gorham, 1991; Turunen et al., 2002; Yu et al., 2010),
amounting to approximately one-fourth of the global soil C pool (2000 – 2700 PgC)
and one-half of the current atmospheric C pool (828 PgC) (Ciais et al., 2013; Jackson
et al., 2017). More than half of this carbon was accumulated before 7000 years ago
during the Holocene, in environments where plant litter production exceeds decay in
water-logged, low-temperature conditions (Yu, 2012). Despite being one of the most
effective ecosystems at sequestering $CO_2$ from the atmosphere over the long-term,
northern peatlands are one of the largest natural sources of methane ($CH_4$), playing a
pivotal role in the global greenhouse gas balance (MacDonald et al., 2006; Mikaloff
Fletcher et al., 2004; Smith, 2004)**.**
The carbon balance of peatlands is sensitive to climate variability and climate change
(Chu et al., 2015; Lund et al., 2012; Yu et al., 2003a). Projected climate warming and
precipitation changes press us to understand the mechanisms of peat growth and
stability, and further to assess the fate of the substantial amount of carbon stored in
peatlands and its potential feedbacks on the climate. Several Land Surface Models
(LSMs) have included representations of the biogeochemical and physical processes of
peatlands to simulate the observed past extent and carbon balance of peatlands and
predict their responses to future climate change (Chaudhary et al., 2017a, 2017b;
Frolking et al., 2010; Kleinen et al., 2012; Spahni et al., 2013; Stocker et al., 2014;
Wania et al., 2009a, 2009b; Wu et al., 2016). The water table depth (WTD) is one of
the most important factors controlling the accumulation of peat, because its position in
the soil column prevents oxygen supply to the saturated zone and reduces





decomposition rates of buried organic matter (Kleinen et al., 2012; Spahni et al., 2013).
It is highlighted by observed and experimental findings, that variations in ecosystem
respiration (ER) depend on WTD (Aurela et al., 2007; Flanagan and Syed, 2011).
However, some studies showed that below a critical level, the drawdown of the water
table did not lead to a significant decrease of soil moisture content, and caused only
small changes in soil air-filled porosity and hence exerted no significant effect on ER
(Lafleur et al., 2005; Parmentier et al., 2009; Sulman et al., 2009). Therefore, while
studying the interactions between peatland water and carbon balances, the dynamics of
soil moisture deserves special attention.
The two-layered (acrotelm-catotelm) conceptual framework was chosen by many
Earth System Models (ESMs) groups to describe peatland structures. The peat profile
was divided into an upper layer with a fluctuating water table (acrotelm) and a lower,
permanently saturated layer (catotelm) – using depth in relation to a drought water table
or a constant value (a widely used depth is 0.3 m below the soil surface) as the discrete
boundary of these two layers (Kleinen et al., 2012; Spahni et al., 2013; Wania et al.,
2009a). This diplotelmic model assumes that all threshold changes in peatland soil
ecological, hydrological and biogeochemical processes occur at the same depth,
causing the lack of generality and flexibility in the model, and thus possibly hindering
the representation of the horizontal and vertical heterogeneity of peatlands (Fan et al.,
2014; Morris et al., 2011).
To our knowledge, only two models attempted to simulate peatland area dynamics
for large-scale gridded applications (Kleinen et al., 2012; Stocker et al., 2014). Kleinen
et al. (2012) modelled wetland extent and peat accumulation in boreal and arctic
peatlands over the past 8000 years using the LPJ model. In their study, simulated
summer mean, maximum and minimum wetland extent by TOPMODEL are used as
surrogates for peatland area, from the assumption that peatland will only initiate and
grow in frequently inundated areas. Stocker et al. (2014) extended the scope of Kleinen
et al. (2012) in the LPX model, distinguishing areas that are suitable for peatland
development using water balance and peatland C balance criteria. While both studies
made pioneering progresses in the modelling of peatland ecosystems, they adopted a





simple bucket approach to model peatland hydrology and peatland C accumulation, and
neither of them resolved the diel cycle of surface energy budget.
To tackle these discrepancies and estimate the C dynamic as well as the peat area, we
used the ORCHIDEE-MICT land surface model incorporating peatland as a sub-grid
hydrological soil unit (HSU). The vertical water fluxes and dynamic carbon profiles in
peatlands are simulated with a multi-layer scheme instead of a bucket model or a
diplotelmic model. A cost-efficient TOPMODEL approach is applied to simulate the
dynamics of peatland area extent. The aim of this study is to model the spatial extent of
northern peatlands since the Holocene and to reproduce peat carbon accumulation over
the Holocene.
**2. Model description**
ORCHIDEE-MICT is an updated version of the ORCHIDEE land surface model with
an improved and evaluated representation of high-latitude processes. Phase changes of
soil water (freeze/thaw), three-layered snowpack and its insulating effects on soil
temperature in winter, permafrost physics and its impacts on plant water availability
and soil carbon profiles are all represented in this model (Guimberteau et al., 2018).
Based on ORCHIDEE-MICT, ORCHIDEE-PEAT is specifically developed to
dynamically simulate northern peatland extent and peat accumulation. ORCHIDEE-
PEAT version 1 was evaluated and calibrated against eddy-covariance measurements
of $CO_2$ and energy fluxes, water table depth, as well as soil temperature from 30
northern peatland sites (Qiu et al., 2018). Parameterizations of peatland vegetation and
water dynamics are unchanged from ORCHIDEE-PEAT version 1: one peatland plant
functional type (PFT) with shallow roots, lateral water flow from surface runoff of non-
peatland areas in the grid cell to peatland, vertical water fluxes in peatland tile with
peat-specific hydraulics (Text S1 in the Supplement). Here, we improve peatland C
dynamics by replacing the diplotelmic peatland C model with a many-layered one. The
32-layered thermal and C models in the standard ORCHIDEE-MICT is used to simulate
peatland C accumulation and decomposition (Sect. 2.1). With fine resolution in the soil
surface (10 layers for the top 1m), this 32-layer model better represents the effects of
soil temperature, soil freezing, and soil moisture on carbon decomposition continuously



within the peat profile than a diplotelmic model. Furthermore, the computationally
efficient TOPMODEL approach proposed by Stocker et al. (2014) is incorporated into
the model to simulate dynamics of peatland area, calibrated with a new dataset of
wetland areas excluding permanent lakes (Sect. 2.2). This model simulating the
dynamics of peatland extent and the vertical buildup of peat is hereinafter referred to as
ORCHIDEE-PEAT v2.0. It is worth mentioning that Guimberteau et al. (2018) defined
soil thermal properties of a specific grid cell as the weighted average of mineral soil
and pure organic soil in that grid, with C content of the grid cell derived from the soil
organic C map from NCSCD and HWSD. This development makes it possible to
include the impacts of peat carbon on the gridcell soil thermics, and is activated in this
study.
**2.1 Modeling peat accumulation and decomposition**
The model has two litter C pools (metabolic and structural) and three soil C pools
(active, slow and passive); all pools are vertically discretized into 32 layers, with
exponentially coarser vertical resolution as depth increases and a total depth of 38 m.
Decomposition of the C in each pool and the C fluxes between the pools are calculated
at each layer, with each pool having a distinct residence time. A detailed description of
the litter and soil C pools and carbon flows between them can be found in the
Supplement Text S2.
**2.1.1   Peat carbon decomposition**
Decomposition of peat soil C is calculated at each layer, controlled by base
decomposition rates of different pools modified by soil temperature, moisture and depth:
$k_{i,l} = k_{0,i} \times f_{T,l} \times f_{M,l} \times f_{Z,l}$   ,                                  (1)
where $k_{i,l}$ is the decomposition rate of the pool $i$ at layer $l$, $k_{0,i}$ is the base
decomposition rate of pool $i$, $f_{T,l}$ is the temperature modifier at layer $l$, $f_{M,l}$ is the
moisture modifier, $f_{Z,l}$ is a depth modifier that further reduces decomposition at depth.
For unfrozen soils, the temperature modifier is an exponential function of soil
temperature, while below 0℃ when liquid water enabling decomposition disappears,
respiration linearly drops to zero at −1℃ (Koven et al., 2011). The soil moisture
modifier is prescribed from the meta-analysis of soil volumetric water content ($m^3 m^{-3}$)



- respiration relationship for organic soils conducted by Moyano et al. (2012). See
Supplement Text S3 for a more detailed description of the temperature and moisture
modifier.
Following Koven et al. (2013), we implement a depth modifier ($f_{Z,l}$) to represent
unresolved depth controls (i.e. priming effects, sorption of organic molecules to mineral
surfaces) on C decomposition. This depth modifier decreases exponentially with depth:
$$f_{Z,l} = \exp\left(-\frac{z_l}{z_0}\right) \ , \tag{2}$$
where $z_l$ (m) is the depth of the layer $l$, $z_0$ (m) is the e-folding depth of base
decomposition rate.

### 2.1.2  Vertical buildup of peat

Water-logging and cold temperature in northern peatland regions prevent complete
decomposition of dead plant material, causing an imbalance between litter production
and decay (Parish et al., 2008). The un-decomposed plant residues accumulate as peat,
and consequently, the peat surface shows an upward growth. Instead of modeling this
upward accumulation of peat, we simulate a downward movement of C by adapting the
method that Jafarov and Schaefer (2016) used to build up a dynamic surface organic
layer.
From 102 peat cores from 73 sites (Lewis et al., 2012; Loisel et al., 2014; McCarter
and Price, 2013; Price et al., 2005; Tfaily et al., 2014; Turunen et al., 2001; Zaccone et
al., 2011), we compiled bulk density (BD) measurements into depth bins which
correspond to the top 17 soil layers (~8.7 m) of the model (Fig. S1a). The median
observed bulk density at each depth bin is assigned to the corresponding soil layer of
the model ($BD_l$). For deeper soil layers of the model (18th - 32th), the value of the 17th
soil layer is used. The fraction of C (% weight) of each soil layer ($\alpha_{cl}$) is derived from
a regression with bulk density from 39 cores from 29 sites (Fig. S1b). With these data,
we calculate the empirical amount of C that each soil layer can hold:
$$M_l = BD_l \times \alpha_{cl} \times \Delta Z_l \ , \tag{3}$$
where $BD_l$ (kg m$^{-3}$) is the soil bulk density of layer $l$, $\alpha_{cl}$ is the mass fraction of
carbon in the soil, and $\Delta Z_l$ (m) is the thickness of the layer.





We then model the vertical downward movement of C between soil layers to mimic
the aggradation of carbon in the peat as follows: If carbon in layer $l$ ($C_l$) exceeds a
maximum amount ($M_{th,l}$), a prescribed fraction ($f$) of the carbon is moved to the layer
below ($l+1$). Here, the carbon flux from layer $l$ to the layer below ($l+1$) is calculated
as:
$$flux_{l \to l+1} = \begin{cases} 0, & C_l < M_{th,l} \\ f \times C_l & C_l \geq M_{th,l} \end{cases} , \tag{4}$$

where $C_l$ (kg m$^{-2}$) is the carbon content of layer $l$. The threshold amount of carbon
in layer $l$ ($M_{th,l}$) is a prescribed fraction ($f_{th}$) of the empirically determined $M_l$:
$$M_{th,l} = f_{th} \times M_l , \tag{5}$$

The values of model parameters $f$ and $f_{th}$ do not change with soil depth.
Finally, the total peat depth is defined as the depth that carbon can be transferred to:
$$H = \frac{C_k}{M_k} \times \Delta Z_k + \sum_{i=1}^{k-1} \Delta Z_i , \tag{6}$$

where $k$ is the deepest soil layer where carbon content is greater than 0, $C_k$ (kg m$^{-2}$)
is the carbon content of layer $k$, $M_k$ (kg m$^{-2}$) is empirical amount of carbon that layer
$k$ can hold, and $\Delta Z_k$ (m) is the thickness of layer $k$.
**2.2 Simulating dynamic peatland area extent**
In grid-based simulations, each grid cell is characterized by fractional coverages of
PFTs. The dynamic coverage of each non-peatland PFT is determined by the DGVM
equations as functions of bioclimatic limitations, sapling establishment, light
competition and natural plant mortality (Krinner et al., 2005; Zhu et al., 2015). Here,
dynamics of peatland area is calculated by a cost-efficient TOPMODEL (Stocker et al.

206 2014).

**2.2.1   The cost-efficient TOPMODEL**
Concepts of TOPMODEL (Beven and Kirkby, 1979) have been proven to be effective
at outlining wetland areas in current state-of-the-art LSMs (Kleinen et al., 2012;
Ringeval et al., 2012; Stocker et al., 2014; Zhang et al., 2016). Based on TOPMODEL,
sub-grid-scale topography information and soil properties of a given watershed / grid
cell are used to redistribute the mean water table depth to delineate the extent of sub-





grid area at maximum soil water content. The empirical relationship between the
flooded fraction of a grid cell and the grid cell mean water table position ($\overline{WT}$) can be
established (Fig. S2a) and approximated by an asymmetric sigmoid function, which is
more computationally efficient (Stocker et al., 2014). Here, we adopted the cost-
efficient TOPMODEL from Stocker et al. (2014) and calibrated TOPMODEL
parameters for each grid cell to match the spatial distribution of northern wetlands (see
more details in Text S4). Tootchi et al. (2018) reconciled multiple current wetland
datasets and generated several high-resolution composite wetland (CW) maps. The one
used here (CW-WTD) was derived by combining regularly flooded wetlands (RFW),
which is obtained by overlapping three open-water and inundation datasets (ESA-CCI
(Herold et al. 2015), GIEMS-D15 (Fluet-Chouinard et al., 2015), and JRC (Fluet-
Chouinard et al., 2015)), with areas that have shallow (WT ≤ 20cm) water tables (Fan
et al., 2013). CW-WTD wetlands are static and aim at representing the climatological
maximum extent of active wetlands and inundation. We therefore compare simulated
monthly maximum wetland extent over 1980−2015 with CW-WTD to calibrate
TOPMODEL parameters. Note that lakes from the HydroLAKES database have been
excluded from the CW-WTD map because of their distinct hydrology and ecology
compared with wetlands (Tootchi et al., 2018).
**2.2.2   Peatland development criteria**
The criteria used to constrain peatland area development are greatly inspired by Stocker
et al. (2014), but with some adaptions.
The initiation of peatland only depends on moisture conditions of the grid cell (Fig.
S2b③ − ⑦): First, only the sub grid cell area fraction that is frequently inundated has
the potential to become peatland ($f_{pot}$). Stocker et al. (2014) determined a 'flooding
persistency' parameter ($N$ in Eq.12, Eq.13 in Stocker et al. (2014)) for the DYPTOP
model by comparing simulated peatland area fraction and total C storage with
observations. $N$ is a globally uniform parameter in DYPTOP, being set to 18 months
during the preceding 31 years. However, the formation of peat is a function of local
climate, and thus suitable formation conditions for peatland vary between geographic
regions. To be specific, the accumulation of peat in arctic and northern latitudes is due



both to high water table and to low temperature, while it is mainly a result of water-
logging conditions in sub-tropical and tropical latitudes (Parish et al., 2008). Therefore,
it is essential to apply different values for the 'flooding persistency' parameter for
different regions, according to local climate conditions. We re-defined the requirement
of persistent flooding for peatland formation as: the area fraction that has the potential
to become peatland needs to be flooded at least *Num* months during the preceding 30
years, with *Num* being the total number of growing season months (monthly air
temperature > 5 °C) in 30 years (Fig. S2b⑤). In this case, with the help of relatively
low air temperature making shorter growing seasons, arctic and boreal latitudes need
shorter inundation periods than sub-tropical and tropical regions to form peatland.
Furthermore, as *Sphagnum*-dominated peatlands are sensitive to summer moisture
conditions (Alexandrov et al., 2016; Gignac et al., 2000), the summer water balance of
the grid cell needs to pass a specific threshold (*SWB*) to form peat and to achieve the
potential peatland area (Fig. S2b⑦). The summer water balance is calculated as the
difference between total precipitation (P) and total potential evapotranspiration (PET)
of May-September. We consider *SWB* as a tunable parameter in the model and run
simulations with $SWB = -6$ cm, 0 cm, 3 cm, and 6 cm. $SWB = 6$ cm is selected so that
the model captures the southern frontier of peatland in Eurasia and western North
America (Text S5).
After the initiation, the development of peatland area is controlled by both moisture
conditions of the grid cell and the long-term carbon balance of the peatland HSU (Fig.
S2c⑨ – ⑰). If the climate becomes drier and the calculated potential peatland area is
smaller than the current peatland area, the peatland HSU area will contract to the new
potential peatland area fraction (Fig. S2c⑫). Otherwise (Fig. S2c⑬), the peatland has
the possibility to expand when the summer water balance threshold is passed. If these
above criteria are satisfied, the final decision depends on the carbon density of the
peatland ($C_{peat}$): the peatland can expand only when long-term input exceeds decay
and a certain amount of C ($C_{lim}$) has accumulated (Fig. S2c⑰). $C_{lim}$ is a product of a
mean measured peat depth (1.07 m) from 40 peat cores (with peat age greater than 1.8
ka but smaller than 2.2 ka) from North American peatland (Gorham et al., 2007, 2012)

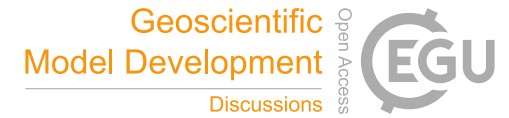



and from the West Siberian lowlands (Kremenetski et al., 2003), a dry bulk density
assumption of 100.0 kg m$^{-3}$ and a mean C fraction of 47% in total peat (Loisel et al.,
2014). Our estimation for $C_{lim}$ is 50.3 kg C m$^{-2}$, matches well with the C density
criterion (50 kg C m$^{-2}$) chosen by Stocker et al. (2014) to represent typical peatland soil.

277        The moisture conditions are evaluated every month throughout the simulation, while

$C_{peat}$ is checked only in the first month after the subC in Spin-up1 and is checked
every month in Spin-up2 and the transient simulation (see Sect. 3.2). The peatland area
fraction ($f_{peat}$) is updated every month. During the simulation, the contracted area and
C are allocated to an 'old peat' pool and are kept track of by the model.
Parameterizations of this pool are identical to mineral soils. When peatland expansion
happens, the peatland will first expand into this 'old peat' area and inherit its stored C.
**3.  Simulation setup and evaluation datasets**
**3.1 Critical Model parameters**
The base decomposition rates of active, slow and passive peat soil carbon pools in the
model are 1.0 a$^{-1}$, 0.027 a$^{-1}$ and 0.0006 a$^{-1}$ at reference temperature of 30 °C,
respectively (Sect. 5). The e-folding depth of the depth modifier ($z_0$, Eq. 2) determines
the general shape of increases of soil C turnover time with depth; the prescribed
threshold to allow downward C transfer between soil layers ($f_{th}$, Eq. 5) and the
prescribed fraction of C to be transferred ($f$, Eq. 4) determine movement and
subsequent distribution of soil C along the soil profile. We compare simulated C vertical
profiles with observed C profiles at 15 northern peatland sites (Table S1) (Loisel et al.,
2014) using different combinations of parameters ($z_0 = (0.5, 1.0, 1.5, 2.0)$, $f_{th} =$
$(0.5, 0.7, 0.9)$ and $f = (0.1, 0.2, 0.3)$) and eventually selected $z_0 = 1.5\,m$, $f_{th} = 0.7$
and $f = 0.1$ based on visual examinations to match the observed C content. Model
sensitivity to the selection will be discussed in Sect. 5.
**3.2 Simulation protocol**
We conduct both site-level and regional simulations with ORCHIDEE-PEAT v2.0 at 1°
× 1° spatial resolution. Regional simulations are performed for the Northern
Hemisphere (>30° N), while site-level simulations are performed for 60 grid cells
containing at least one peat core (Table S1, Fig. S3). Peat cores used in site-level



simulations are from the Holocene Perspective on Peatland Biogeochemistry database
(HPPB) (Loisel et al., 2014). Both site-level and regional simulations are forced by the
6-hourly meteorological forcing from the CRUNCEP v8 dataset, which is a
combination of the CRU TS monthly climate dataset and NCEP reanalysis
(https://vesg.ipsl.upmc.fr/thredds/catalog/store/p529viov/cruncep/V7_1901_2015/cata
log.html).

309        All simulations start with a two-step spin-up followed by a transient simulation after

the pre-industrial period (Fig. S4). The first spin-up (Spin-up1) includes $N$ cycles of a
peat carbon accumulation acceleration procedure consisting of 1) 30 years with the full
ORCHIDEE-PEAT (FullO) run on 30 min time step followed by 2) a stand-alone soil
carbon sub-model (SubC) run to simulates the soil carbon dynamics in a cost effectively
way on monthly steps (fixed monthly litter input, soil water and soil thermal conditions
from the preceding FullO simulation). Repeated 1961−1990 climate forcing is used in
Spin-up1 to approximate the higher Holocene temperatures relative to the preindustrial
period (Marcott et al., 2013). The atmospheric $CO_2$ concentration is fixed at the
preindustrial level (286 ppm). Each time we run the SubC for 2000 years (2 ka) in the
first $N-1$ sets of acceleration procedures while, the value of $N$ and the time length of
the last set of acceleration procedure ($X$) are defined according to the age of the peat
core in site-level simulations, and are defined according to the reconstructed glacial
retreat in regional simulations (Fig. S5, S6). The reconstructed glacial retreat used in
this study are from Dyke (2004) for North America and are from Hughes et al. (2016)
for Eurasia (Text S6).

325        In the second spin-up step (Spin-up2), the full ORCHIDEE-PEAT model was run for

100 years, forced by looped 1901−1920 climate forcing and preindustrial atmospheric
$CO_2$ concentration so that physical and carbon fluxes can approach to the preindustrial
equilibrium. After the two spin-ups, a transient simulation is run, forced by historical
climate forcing from CRUNCEP and rising atmospheric $CO_2$ concentration. For site-
level simulations, the transient period starts from 1860 and ends at the year of coring
(Table S1). For regional simulations, the transient period starts from 1860 and ends at

332    2009.





### 3.3 Evaluation datasets

### 3.3.1 Evaluation datasets for site-level simulations

All peatland sites used in this study are from the HPPB database (Loisel et al., 2014). All the peat cores measured peat ages and depths (60 sites, Table S1), hence are used to evaluate simulated peat depth, with sites being grouped into different peatland types, climate zones and ages. For peat cores where peat ages, depths, fraction of C and bulk density were recorded (15 sites marked in red in Table S1), we construct vertical C profiles with this measured information to compare with our simulated C profiles.

### 3.3.2 Northern peatland evaluation datasets for regional simulations

**Area**

Simulated peatlands area in 2009 is evaluated against: 1. World Inventory of Soil Emission potentials (WISE) database (Batjes, 2016); 2. An improved global peatland map (PEATMAP) by reviewing a wide variety of global, regional and local scale peatland distribution information (Xu et al., 2018); 3. International Mire Conservation Group Global Peatland Database (IMCG-GPD) (Joosten, 2010); 4. Peatland distribution map by Yu et al. (2010).

**Soil organic carbon stocks**

Simulated peatlands SOC is evaluated against: 1. The WISE database; 2. The IMCG-GPD.

All the above-mentioned datasets used to evaluate ORCHIDEE-PEAT v2.0 at regional scale are described in the Supplement Text S7.

**Peat depth**

Gorham et al. (2007, 2012) and Kremenetski et al. (2003) collected depth and age of 1685 and 130 peat cores, respectively, from literature data on peatlands in North America (NA) and in the West Siberian lowlands (WSL). These compilations make it possible for us to validate peat depths simulated by ORCHIDEE-PEAT v2.0 at regional scales, in addition to the detailed site-runs in Sect. 3.3.1. Compared to the HPPB database, these datasets lack detailed peat properties (i.e. C content, peatland type…), but contain more samples and cover larger areas.

### 4. Results





**4.1 Site simulation**


We first evaluate the performance of ORCHIDEE-PEAT v2.0 in reproducing peat
depths and vertical C profiles at the 60 sites from HPPB (Table S1). Out of the 60 grid
cells (each grid cell corresponding to one peat core), ORCHIDEE-PEAT v2.0 produces
peatlands in 57 of them. The establishment of peatlands at Zoige, Altay and IN-BG-1
(Table S1) is prevented in the model by the unmet water balance criteria of these grid
cells. Simulated peat depth of these 57 sites ranges from 0.37 m to 6.64 m and shows a
median depth of 2.18m (Table 1), shallower than observations (ranges from 0.96 to
10.95 m, with the observed median depth being 3.10 m). The root mean square error
(RMSE) between observations and simulations is 2.45 m.
The measured and simulated median peat depths for the 14 fen sites are 3.78 m and
2.16 m, compared to 3.30 m and 2.18 m, respectively for the 33 bog sites (Table 1). The
model shows slightly higher accuracy for fens than for bogs, with RMSE for fens being
2.08 m and 2.59 m for bogs (Fig. 1a). RMSE for peat depths of sites that are older than
8 ka are greater than that of younger sites, but are smaller than the measured mean depth
(3.5 m) of all peat cores (Fig. 1b). Simulated peats are deeper than observations at the
6 arctic sites, but are shallower than observations at the 47 boreal sites and at the 4
temperate sites (Table 1). The RMSE for temperate sites rises above the measured mean
depth of all cores (Fig. 1c).
The simulated and observed vertical profiles of soil C for the 15 sites are shown in
Fig. 2, simulated C concentrations are generally within the range of measurements at
most of the sites, but are underestimated at Sidney bog, Usnsk Mire 1, Lake 785 and
Lake 396. In the model, the buildup of peat is parameterized by downward movement
of C between soil layers, with the maximum amount of C that each layer can hold being
calculated from median observed bulk density and C fraction of peat core samples (Sect.
2.1.2). High C concentration of cores that have significantly larger bulk density and /
or C fraction than the median of the measurements thus cannot be reproduced. This is
the case of Lake 785 and Lake 396 (Table S1), where C concentrations are
underestimated and depths are overestimated (Fig. 2), while simulated total C content
is close to observations (for Lake 785, measured and simulated C content is 86.14





393 kgC m$^{-2}$ and 96.13 kgC m$^{-2}$, respectively, while values for Lake 396 are 57.2 and

394 70.2 kgC m$^{-2}$).

395  As shown in Fig. 3, there is considerable variability in depth and C concentration

396 profiles among peat cores within a grid cell, even though these cores have a similar age.

397 We rerun the model at the 5 grid cells where more than one peat core has been sampled,

398 with time length of the simulation being defined as the mean age of cores in the same

399 one grid cell. The simulated peat depth and C concentration profiles at G2, G4, and G5

400 are generally within the range of peat core measurements (Fig. 3). G1 and G3 is the

401 same case as Lake 785 and Lake 396.

402 **4.2 Regional simulation**

403 **Northern peatlands area and C stock**

404 Simulated maximum inundated area of the Northern Hemisphere is 9.1 million km$^2$,

405 smaller than the wetland areas in CW-WTD (~13.2 million km$^2$ after excluding lakes).

406 TOPMODEL gives an area fraction at maximum soil water content while CW-WTD

407 includes both areas seasonally to permanently flooded and areas that are persistently

408 saturated or near-saturated (the maximum water table shallower than 20 cm) soil-

409 surface. Therefore, an exact match between CW-WTD and the model prediction is not

410 expected. The model generally captures the spatial pattern of wetland areas represented

411 by CW-WTD (Fig. S7).

412  While our model predicts the natural extent of peatlands under suitable climate

413 conditions, soil formation processes and soil erosion are not included in the model. We

414 mask grid cells that are dominated by Leptosols, which are shallow or stony soils over

415 hard rock, or highly calcareous material (Nachtergaele, 2010) (Fig. S8, Fig. S9).

416 Peatlands have been extensively used for agriculture after drainage and / or partial

417 extraction worldwide (Carlson et al., 2016; Joosten, 2010; Leifeld and Menichetti, 2018;

418 Parish et al., 2008). Intensive cultivation practices might cause rapid loss of peat C and

419 ensuing disappearance of peatland. Additionally, agricultural peatlands are often

420 classified as cropland, not as organic soils (Joosten, 2010). Therefore, we masked

421 agricultural peatland from the results by assuming that crops occupy peatland in

422 proportion to the grid cell peatland area (Carlson et al., 2016). The distribution and area





of cropland used here is from the MIRCA2000 data set (Portmann et al., 2010), which
provides monthly crop areas for 26 crop classes around the year 2000 and includes
multicropping explicitly (Fig. S10). After masking Leptosols and agricultural peatlands,
the simulated total northern peatlands area is 3.9 million km$^2$ ($f_{\text{noLEP-CR}}$, Fig. 4d),
holding 463 PgC ($C_{\text{noLEP-CR}}$, Fig. 5b). These estimates fall well within estimated ranges
of northern peatland area (3.4 – 4 million km$^2$ ) and carbon stock (270 – 540 PgC)
(Gorham, 1991; Turunen et al., 2002; Yu et al., 2010). Simulated peatland area matches
relatively well with PEATMAP data in Asian Russia but overestimates peat area in
European Russia (Table 2). The simulated total peatlands area of Canada is in relatively
good agreement with the three evaluation data sets, though the hotspot at the Hudson
Bay lowlands is underestimated and a small part of the northwest Canada peatlands is
missing. In Alaska, the simulated distribution of peatland area agrees well with Yu et al.
(2010) and WISE. There is a large overestimation of peatland area in southeastern US
(Table 2, Fig. 4d). The simulated peat C stock in Russia (both the Asian and the
European part), and in US are overestimated compared to IMCG-GPD and WISE, but
that of Canada is underestimated (Table 3, Fig. 5b).
**Peat depth**
Fig. 6 shows measured and simulated peat depth in NA and WSL. Some peat cores are
sampled from the Canadian Arctic Archipelago, southwestern US and the northern tip
of Quebec, where there is no peatland in peat inventories / the soil database. These sites
support the notion that the formation and development of peatland are strongly
dependent on local conditions, i.e. retreat of glaciers, topography, drainage, vegetation
succession (Carrara et al., 1991; Madole, 1976). We do not expect the model to capture
every single peatland because it is a large-scale LSM. Therefore, cores that are not
captured by the model are removed from further analysis.
As shown in Fig. 3, within a grid cell, sampled peat cores can have very different
depths and / or ages. We calculate the mean depth of cores in each of the grid cells and
compare it against the simulated mean depth. The mean age of cores in each of the grid
cells is used to determine which output of the model should be examined. For instance,
the mean age of the four cores in grid cell (40.5 °N, 74.5 °W) is 2.5 ka, and accordingly,



we pick out the simulated depth of this grid cell right after the first run of SubC (Fig.
S4) to compare with the mean depth of these cores. We acknowledge that this is still a
crude comparison since the simulation protocol implies that we can only make the
comparison at 2000-year intervals. Nonetheless, it is a compromise between running
the model for 1815 peat cores independently and comparing the mean depth of
measured points with grid-based simulated depth. As shown in Fig. 7, for each age
interval (of both the West Siberian lowlands and North America), the variation in
simulated depth is smaller than that in the measurement. The two deepest simulated
peat in WSL belong to the fourth age group ($6 < \text{Age} \leq 8$ ka) and are the result of a
shallow active layer; while C is moving downward to deeper and deeper layers, the
decomposition is greatly limited by cold conditions at depth. At both WSL and NA,
simulated median peat depths ($2.07 – 2.36$ m at WSL, $1.02 – 2.15$ m at NA ) are in
relatively good agreement with measurements ($1.8 – 2.31$ m at WSL, $0.8 – 2.46$ m at
NA) for cores younger than 8 ka (Fig. 7). For the two oldest groups (peat age > 8 ka),
the simulated median depths are about 0.70 m shallower than measurements at NA and
about 1.04 m shallower at WSL.

**469   Undisturbed northern peatland carbon balance in the past century**

Simulated mean annual (averaged over $1901 – 2009$) net ecosystem production (NEP)
of northern peatlands varies from $– 63$ gC m$^{-2}$ a$^{-1}$ to 46 gC m$^{-2}$ a$^{-1}$ (Fig. 8). The West
Siberian lowlands, the Hudson Bay lowlands, Alaska, and the China-Russia border are
significant hotspots of peatland C uptake. Simulated mean annual NEP of all northern
peatlands over $1901 – 2009$ is 0.1 PgC a$^{-1}$, consistent with the previous estimate of
0.076 PgC a$^{-1}$ by Gorham (1991) and the estimate of 0.07 PgC a$^{-1}$ by Clymo et al.
(1998). From 1901 to 2009, both net primary production (NPP) and heterotrophic
respiration (HR) show an increasing trend, but NPP rises faster than HR during the
second half of the century (Fig. 9a). The increase of NPP is caused by atmospheric $CO_2$
concentration and increasing of air temperature (Fig. 9, Fig. S11). As air (soil)
temperature increases, HR also increases but lags NPP (Fig. 9, Fig. S11). Simulated
annual NEP ranges from $–0.03$ PgC a$^{-1}$ to 0.23 PgC a$^{-1}$, with a significant positive trend
over the second half of the century (Fig. 9b). NEP shows a significant positive





relationship with air (soil) temperature and with atmospheric $CO_2$ concentration (Fig.
S11). $CH_4$ and dissolved organic carbon (DOC) are not yet included in the model, both
of them are significant losses of C from peatland (Roulet et al., 2007).
**5. Discussion**
**Peat depth**
We found a general underestimation of peat depth (Fig. 1, Fig. 7), possibly due to the
following several reasons. Firstly, there is a lack of specific local climatic and
topographic conditions: The surfaces of peatlands are mosaics of microforms, with
accumulation of peat occurring at each individual microsites of hummocks, lawns and
hollows. Differences in vegetation communities, thickness of the unsaturated zone,
local peat hydraulic conductivity and transmissivity between microforms result in
considerable variation in peat formation rate and total C mass (Belyea and Clymo, 2001;
Belyea and Malmer, 2004; Borren et al., 2004; Packalen et al., 2016). Cresto Aleina et
al. (2015) found that the inclusion of microtopography in the Hummock-Hollow model
delayed the simulated runoff and maintained wetter peat soil for a longer time at a
peatland of Northwest Russia, thus contributed to enhanced anoxic conditions.
Secondly, site-specific parameters are not included in gridded simulations: Parameters
describing peat soil properties, i.e., soil bulk density and soil carbon fraction, determine
the amount of C that can be stored across the vertical soil profile. Hydrological
parameters, i.e., the hydraulic conductivity and diffusivity, and the saturated and
residual water content, regulate vertical fluxes of water in the peatland soil and
expansion/contraction of the peatland area, and hence influence the decomposition and
accumulation of C at the sites considered. Plant trait parameters, i.e. the maximal rate
of carboxylation ($V_{cmax}$), the light saturation rate of electron transport ($J_{max}$) determine
the carbon budgets of the sites (Qiu et al., 2018). The depth modifier, which
parameterizes depth dependence of decomposition, controls C decomposition at depth
and is an important control on simulated total C and the vertical C profile. A third reason
is sample selection bias: Ecologists and geochemists tend to take samples from the
deepest part of a peatland complex to obtain the longest possible records (Gorham, 1991;
Kuhry and Turunen, 2006). In contrast, the model is designed to model an average age



and C stock of peatlands in a grid location and thus preferably, the simulated C
concentrations of a grid cell should only be validated against grids represented by a
number of observed cores. We do try to compare the model output with multiple peat
cores (Fig. 3, Fig. 7), but shallow peats are not sufficiently represented in field
measurements. A fourth source of error is that simulated initiation time of peat
development at some sites are too late compared to ages of measured cores: The model
multiple spin-up strategy is designed to account for coarse-scale ice-sheet distribution
at discrete Holocene intervals (Sect. 3.2, Fig. S4), and if the modelled occurrence of
peatland is too late, the accumulated soil C may be underestimated. For example, at the
Patuanak site, where the core age is 9017 ka, the model was run with 4 times' SubC
(Table S1). However, there was no peatland before the first SubC, meaning that
simulated peatland at this grid cell was 2000 years younger than the observation and
that our simulation missed C accumulation during the first 2000 years at this site. This
may be another source of bias associated with the model resolution, namely that local
site conditions fulfilled the initiation of peatland at specific locations, but the average
topographic and climatic conditions of the coarse model grid cell were not suitable for
peatland initiation. Also, one has to keep in mind that a single / a few sample (s) from
a large peat complex may not be enough to capture the lateral spread of peat area, which
may be an important control on accumulation of C (Charmen, 1992; Gallego-Sala et al.,
2016; Parish et al., 2008). The underestimation of peat depth can also come from biased
climate input data: Spin-ups of the model are forced with repeated 1961−1990 climate,
assuming that Holocene climate is equal to recent climate. While peatland carbon
sequestration rates are sensitive to climatic fluctuations, centennial to millennial scale
climate variability, i.e. cooling during the Younger Dryas period and the Little Ice Age
period, warming during the Bølling-Allerød period are not included in the climate
forcing data (Yu et al., 2003a, 2003b). An early Holocene carbon accumulation peak
was found during the Holocene Thermal Maximum when the climate was warmer than
present (Loisel et al., 2014; Yu et al., 2009). Finally, effects of landscape morphology
on drainage as well as drainage of glacial lakes are not incorporated and can represent
a source of uncertainty.





### Simulated peatland area development

The initiation and development of peatlands in NA followed the retreat of the ice sheets, as a result of the continuing emergence of new land with the potential to become suitable for peatland formation (Gorham et al., 2007; Halsey et al., 2000). To take glacial extent into account for simulating the Holocene development of peatlands, we use ice sheet reconstructions in NA and Eurasia (Fig. S5, S6). Not surprisingly, when ice cover is considered, the area of peatlands that developed before 8 ka is significantly decreased, while the area that developed after 6 ka is increased (Fig. 10). We use observed frequency distribution of peat basal age from MacDonald et al. (2006) as a proxy of peatland area change over time, following the assumption proposed by Yu (2011) that peatland area increases linearly with the rate of peat initiation. We grouped the data of MacDonald et al. (2006) into 2000-years bins to compare with simulated peatlands area dynamics (Fig. 10). The inclusion of dynamic ice sheet coverage triggering peat inception clearly improved the model performance in replicating peatland area development during the Holocene, though the peatland area before 8 ka is still overestimated by the model in comparison with the observed frequency distribution of basal ages (Fig. 10). In spite of the difference in peatlands area expansion dynamics between the simulation that considered dynamic ice sheets and the one that did not, the model estimates of present-day total peatland area and carbon stock are generally similar (Fig. S12). Without dynamic ice sheet, the model would predict only 0.1 million km$^2$ more peatland area and 24 Pg more peat C over the Northern Hemisphere (>30 °N). We are aware of two studies that attempted to account for the presence of ice sheets during the Holocene (Kleinen et al., 2012) and the last Glacial Maximum (Spahni et al., 2013) while simulating peatland C dynamics. Kleinen et al. (2012) modelled C accumulation over the past 8000 years in the peatland areas north of 40 °N using the coupled climate carbon cycle model CLIMBER2-LPJ. A decrease of 10 PgC was found when ice sheet extent at 8 ka BP (from the ICE-5G model) was accounted for. Another peatland modelling study conducted by Spahni et al. (2013) with LPX also prescribed ice sheets and land area from the ICE-5G ice-sheet reconstruction (Peltier, 2004), but influences of ice sheet margin fluctuations on simulated peatland





area and C accumulation were not explicitly assessed in their study.
The peatland carbon density criterion for peatland expansion ($C_{lim}$) is an important
factor impacting the simulated Holocene trajectory of peatlands development. Without
the limitation of $C_{lim}$, a larger expansion of northern peatlands would occur before 10
ka (Fig. S13). Such a premature, 'explosive' increase of peatland area would result into
the overestimation of C accumulated in the early Holocene in the model. In the
meantime, peatland area in regions that only have small C input, i.e. Baffin Island, and
northeast Russia, would be overestimated (Fig. S14).
**Choice of model parameters**
For the active, slow and passive peat soil carbon pool, the base decomposition rates
are 1.0 $a^{-1}$, 0.027 $a^{-1}$ and 0.0006 $a^{-1}$ at reference temperature of 30 °C, respectively,
meaning that the residence times at 10 °C (no moisture and depth limitation) of these
three pools are 4 years, 148 years and 6470 years. In equilibrium / near- equilibrium
state, simulated C in the active pool takes up only a small fraction of the total peat C,
while generally 40% − 80% of simulated peat C are in the slow C pool and about 20%
− 60% are in the passive C pool. Assuming that in a peatland, the active, slow and
passive pool account for 3%, 60%, and 37% (median values from the model output of
the year 2009) of the total peat C, we can get a mean peat C residence time of 2500
years. If depth modifier is considered, the C residence time will vary from 2500 years
at the soil surface to 13200 years at the 2.5 m depth. For the record, in previous
published large-scale diplotelmic peatland models, at 10 °C, C residence time for the
acrotelm (depth = 0.3 m) ranged from 10 to 33 years and ranged from 1000 to 30000
years for the catotelm (Kleinen et al., 2012; Spahni et al., 2013; Wania et al., 2009b).
We performed sensitivity tests to show the sensitivity of the modelled peat C to model
parameters at the 15 northern peatland sites where observed vertical C profiles can be
constructed (Table S1). Tested parameters are the e-folding decreasing depth of the
depth modifier ($z_0$, Eq. 2), the prescribed thresholds to start C transfer between soil
layers ($f_{th}$, Eq. 5) and the prescribed fraction of C transferred vertically ($f$, Eq. 4). We
found that changing $f_{th}$ or $f$ leads to only small effects on the vertical soil C profile
(see e.g. Burnt Village peat site in Fig. S15). The parameter $z_0$, by contrast, exerts a





relatively strong control over C profiles. With smaller $z_0$, decomposition of C
decreases rapidly with depth, resulting in deeper C profile (Fig. S15). Regional scale
tests verified these behaviors of the model: when $f_{th} = 0.9$ is used (instead of $f_{th} =$
0.7), changes in peatland area and peat C stock are negligible (Fig. S16); If $z_0 = 0.5$ m
is applied (instead of $z_0 = 1.5$ m), the simulated total peat C would triple while the
total peatland area would only increase by 0.2 million km$^2$ (Fig. S17). This illustrates
the importance of constraining decomposition rates at depth in peatland models.
**Uncertainties in peatland area and soil C estimations**
There are large uncertainties in estimates of peatland distribution and C storage.
Some studies prescribe peatlands from wetlands. However, in spite of the fact that there
are extensive disagreements between wetland maps, it is a challenge to distinguish
peatlands from non-peat forming wetlands (Gumbricht et al., 2017; Kleinen et al., 2012;
Melton et al., 2013; Xu et al., 2018). Estimates based on peatland inventories are
impeded by poor availability of data, non-uniform definitions of peatlands among
regions and coarse resolutions (Joosten, 2010; Yu et al., 2010). In addition, as peatlands
are normally defined as waterlogged ecosystems with a minimum peat depth of 30 cm
or 40 cm, shallow peats are underrepresented. Another approach to estimate peatland
area and peat C is to use a soil organic matter map to outline organic-rich areas, such
as histosols and histels (Köchy et al., 2015; Spahni et al., 2013). This approach
overlooks local hydrological conditions and vegetation composition (Wu et al., 2017).
Our model estimates of peatland area and C stock generally fall well within the range
of published estimates, except in southeastern US, where there is only 0.05 – 0.10
million km$^2$ of peatland in observations but 0.37 million km$^2$ in the model prediction
(Fig. 4d, Table 2). We notice a large interannual variability in peatland area and C
predictions in southeastern US (Fig. S18), which suggests that some areas are not
suitable for long-term development of peatlands. Another factor that might have
contributed to the overestimation is a limitation of TOPMODEL, namely that the
'floodability' of a pixel in the model is determined by its compound topographic index
(CTI) value regardless of the pixel's location along the stream, and thus the floodability
of an upstream pixel with a large CTI might be affected by downstream pixels that have





small CTI. The model's inability to resolve small-scale streamflows might be another
cause of the overestimation. Fires, historical peat extraction and drainage posed great
dangers to peatlands, but are not considered in this study (Hatala et al., 2012; Turetsky
et al., 2004, 2015).
The simulated mean annual NPP, HR and NEP of northern peatlands increase from
about 1950 onwards. We find positive relationships between NPP and temperature, NPP
and atmospheric $CO_2$ concentration, as well as HR and temperature over the past
century (Fig. S11). From a future perspective, it is unclear whether the increasing trend
of NEP can be maintained. While photosynthetic sensitivity to $CO_2$ decreases with
increasing atmospheric $CO_2$ concentration and photosynthesis may finally reach a
saturation point in the future, decomposition is not limited by $CO_2$ concentration and
may continue to increase with increasing temperature (Kirschbaum, 1994; Wania et al.,
2009b).
Our model applies a multi-layer approach to simulate process-based vertical water
fluxes and dynamic C profiles of northern peatlands, highlights the vertical
heterogeneities in the peat profile in comparison to previous diplotelmic models
(Kleinen et al., 2012; Spahni et al., 2013; Stocker et al., 2014; Wania et al., 2009b).
While simulating peatland dynamics, large-scale models used a static peatland
distribution map obtained from peat inventories / soil classification map (Largeron et
al., 2018; Wania et al., 2009a, 2009b), or prescribed the trajectory of peatland area
development over time (Spahni et al., 2013), or used wetland area dynamics as a proxy
(Kleinen et al., 2012). DYPTOP, however, predicted peatland area dynamics by
combing simulated inundation and a set of peatland expansion criteria (Stocker et al.,
2014). We add the scheme of DYPTOP into our model with some adaptions to simulate
spatial and temporal dynamics of northern peatland area. Further work to improve this
simulation framework is needed in areas such as an accurate representation of the
Holocene climate, higher spatial resolution, distinguish bogs from fens to better
parameterize water inflows into peatland. Including $CH_4$ emissions and leaching of
DOC will be helpful to get a more complete picture of peatland C budget.



## 6. Conclusions


Multi-layer schemes have been proven to be superior to simple box configurations in
ESMs at realistic modeling of energy, water and carbon fluxes over multilayer
ecosystems (De Rosnay et al., 2000; Jenkinson & K. Coleman, 2008; Best et al., 2011;
Wu et al., 2016). We apply multi-layer approaches to model vertical profiles of water
fluxes and vertical C profiles of northern peatlands. Besides representations of peatland
hydrology, peat C decomposition and accumulation, a dynamic model of peatland
extent is also included. The model shows good performance at simulating average peat
depth and vertical C profile in grid-based simulations. Modern total northern peatland
area and C stock is simulated as 3.9 million $km^2$ and 463 PgC (Leptosols and
agricultural peatlands have been marsked), respectively. While this study investigated
the capability of ORCHIDEE-PEAT v2.0 to hindcast the past, in ongoing work, the
model is being used to explore how peatlands area and C cycling may change under
future climate scenarios.




















Author contribution:

CQ implemented peatland water and carbon processes into ORCHIDEE-MICT, introduced the dynamic peatland area module and performed the simulation. DZ contributed to ensuring consistency between the peatland modules and various other processes and modules in the model. PC conceived the project. PC, BG, GK, DZ and CQ contributed to improving the research and interpreting results. SP assisted in implementing of the cost-efficient TOPMODEL. AT and AD provided the dataset of wetland areas. SP, AT, AD and AH contributed to the calibration of the TOPMODEL. All authors contributed to the manuscript.

Code availability:

The source code is available online via: http://forge.ipsl.jussieu.fr/orchidee/browser/branches/publications/ORCHIDEE-PEAT_r5488, Readers interested in running the model should follow the instructions at http://orchidee.ipsl.fr/index.php/you-orchidee.

Acknowledgements:

This study was supported by the European Research Council Synergy grant ERC-2013-SyG-610028 IMBALANCE-P. AH received financial support from the European Union's Horizon 2020 research and innovation program under the Marie Sklodowska-Curie grant agreement No. 643052 (C-CASCADES project). We thank Zicheng Yu for providing the peatland distribution map.

Conflict of Interest:    The authors declare no conflict of interest.

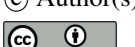



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

Concentration - a Theoretical-Analysis of Its Dependence on Temperature and
Background    CO2    Concentration,    Plant,    Cell    Environ.,    17(6),    747–754,
doi:10.1111/j.1365-3040.1994.tb00167.x, 1994.
Kleinen, T., Brovkin, V. and Schuldt, R. J.: A dynamic model of wetland extent and
peat accumulation: Results for the Holocene, Biogeosciences, 9(1), 235–248,
doi:10.5194/bg-9-235-2012, 2012.
Köchy, M., Hiederer, R. and Freibauer, A.: Global distribution of soil organic carbon –
Part 1 : Masses and frequency distributions of SOC stocks for the tropics , permafrost
regions , wetlands , and the world, Soil, 1(1), 351–365, doi:10.5194/soil-1-351-2015,
841    2015.
Koven, C. D., Ringeval, B., Friedlingstein, P., Ciais, P., Cadule, P., Khvorostyanov, D.,
Krinner, G. and Tarnocai, C.: Permafrost carbon-climate feedbacks accelerate global
warming, Proc. Natl. Acad. Sci., 108(36), 14769–14774, 2011.
Koven, C. D., Riley, W. J., Subin, Z. M., Tang, J. Y., Torn, M. S., Collins, W. D.,
Bonan, G. B., Lawrence, D. M. and Swenson, S. C.: The effect of vertically resolved
soil biogeochemistry and alternate soil C and N models on C dynamics of CLM4,
Biogeosciences, 10(11), 7109–7131, doi:10.5194/bg-10-7109-2013, 2013.



Kremenetski, K. V., Velichko, A. A., Borisova, O. K., MacDonald, G. M., Smith, L.
C., Frey, K. E. and Orlova, L. A.: Peatlands of the Western Siberian lowlands: Current
knowledge on zonation, carbon content and Late Quaternary history, Quat. Sci. Rev.,
22(5–7), 703–723, doi:10.1016/S0277-3791(02)00196-8, 2003.
Krinner, G., Viovy, N., de Noblet-Ducoudré, N., Ogée, J., Polcher, J., Friedlingstein,
P., Ciais, P., Sitch, S. and Prentice, I. C.: A dynamic global vegetation model for studies
of the coupled atmosphere-biosphere system, Global Biogeochem. Cycles, 19(1), 1–33,
doi:10.1029/2003GB002199, 2005.
Kuhry, P. and Turunen, J.: The postglacial development of boreal and subarctic
peatlands, in Boreal peatland ecosystems, pp. 25–46, Springer., 2006.
Lafleur, P. M., Moore, T. R., Roulet, N. T. and Frolking, S.: Ecosystem respiration in
a cool temperate bog depends on peat temperature but not water table, Ecosystems, 8(6),
619–629, doi:10.1007/s10021-003-0131-2, 2005.
Largeron, C., Krinner, G., Ciais, P. and Brutel-Vuilmet, C.: Implementing northern
peatlands in a global land surface model: description and evaluation in the ORCHIDEE
high-latitude version model (ORC-HL-PEAT), Geosci. Model Dev., 11(8), 3279–3297,
865  2018.

Leifeld, J. and Menichetti, L.: The underappreciated potential of peatlands in global
climate change mitigation strategies, Nat. Commun., 9(1), 1071, 2018.
Lewis, C., Albertson, J., Xu, X. and Kiely, G.: Spatial variability of hydraulic
conductivity and bulk density along a blanket peatland hillslope, Hydrol. Process.,
26(10), 1527–1537, doi:10.1002/hyp.8252, 2012.
Loisel, J., Yu, Z., Beilman, D. W., Camill, P., Alm, J., Amesbury, M. J., Anderson, D.,
Andersson, S., Bochicchio, C., Barber, K., Belyea, L. R., Bunbury, J., Chambers, F. M.,
Charman, D. J., De Vleeschouwer, F., Fia kiewicz-Kozie , B., Finkelstein, S. a., Ga ka,
M., Garneau, M., Hammarlund, D., Hinchcliffe, W., Holmquist, J., Hughes, P., Jones,
M. C., Klein, E. S., Kokfelt, U., Korhola,  a., Kuhry, P., Lamarre,  a., Lamentowicz,
M., Large, D., Lavoie, M., MacDonald, G., Magnan, G., Makila, M., Mallon, G.,
Mathijssen, P., Mauquoy, D., McCarroll, J., Moore, T. R., Nichols, J., O'Reilly, B.,
Oksanen, P., Packalen, M., Peteet, D., Richard, P. J., Robinson, S., Ronkainen, T.,
Rundgren, M., Sannel,  a. B. K., Tarnocai, C., Thom, T., Tuittila, E.-S., Turetsky, M.,
Valiranta, M., van der Linden, M., van Geel, B., van Bellen, S., Vitt, D., Zhao, Y. and
Zhou, W.: A database and synthesis of northern peatland soil properties and Holocene
carbon   and   nitrogen   accumulation,   The   Holocene,   24(9),   1028–1042,
doi:10.1177/0959683614538073, 2014.
Lund, M., Christensen, T. R., Lindroth, A. and Schubert, P.: Effects of drought
conditions on the carbon dioxide dynamics in a temperate peatland, Environ. Res. Lett.,
886  7(4), 45704, 2012.

Macdonald, G. M., Beilman, D. W., Kremenetski, K. V, Sheng, Y., Smith, L. C. and
Velichko, A. a: Rapid early development of circumarctic peatlands and atmospheric
CH4 and CO2 variations., Science, 314(5797), 285–288, doi:10.1126/science.1131722,
890  2006.

MacDonald, G. M., Beilman, D. W., Kremenetski, K. V., Sheng, Y., Smith, L. C. and
Velichko, A. A.: Rapid Early Development of Circumarctic Peatlands and Atmospheric



CH4 and CO2 Variations, Science, 314(5797), 285–288, doi:10.1126/science.1131722,
894    2006.

Madole, R. F.: Bog Stratigraphy, Radiocarbon-Dates, and Pinedale to Holocene Glacial
History in Front Range, Colorado, J. Res. Us Geol. Surv., 4(2), 163–169 [online]
Available from: isi:A1976BR69000004, 1976.
Marcott, S. a., Shakun, J. D., Clark, P. U. and Mix, A. C.: A Reconstruction of Regional
and Global Temperature for the Past 11,300 Years, Science, 339(6124), 1198–1201,
doi:10.1126/science.1228026, 2013.
McCarter, C. P. R. and Price, J. S.: The hydrology of the Bois-des-Bel bog peatland
restoration: 10 years post-restoration, Ecol. Eng., 55, 73–81,
doi:10.1016/j.ecoleng.2013.02.003, 2013.
Mcgrath, M. J., Ryder, J., Pinty, B., Otto, J., Naudts, K., Valade, A., Chen, Y.,
Weedon, J. and Luyssaert, S.: A multi-level canopy radiative transfer scheme for
ORCHIDEE ( SVN r2566 ), based on a domain-averaged structure factor, Geosci.
Model Dev. Discuss., 1–22, doi:10.5194/gmd-2016-280, 2016.
Melton, J. R., Wania, R., Hodson, E. L., Poulter, B., Ringeval, B., Spahni, R., Bohn, T.,
Avis, C. A., Beerling, D. J. and Chen, G.: Present state of global wetland extent and
wetland methane modelling: conclusions from a model intercomparison project
(WETCHIMP), Biogeosciences, 10, 753–788, 2013.
Mikaloff Fletcher, S. E., Tans, P. P., Bruhwiler, L. M., Miller, J. B. and Heimann, M.:
CH4 sources estimated from atmospheric observations of CH4 and its 13C/12C isotopic
ratios: 1. Inverse modeling of source processes, Global Biogeochem. Cycles, 18(4),
915    2004.

Morris, P. J., Waddington, J. M., Benscoter, B. W. and Turetsky, M. R.: Conceptual
frameworks in peatland ecohydrology: looking beyond the two-layered (acrotelm–
catotelm) model, Ecohydrology, 4(1), 1–11, 2011.
Moyano, F. E., Vasilyeva, N., Bouckaert, L., Cook, F., Craine, J., Curiel Yuste, J., Don,
A., Epron, D., Formanek, P., Franzluebbers, A., Ilstedt, U., Kätterer, T., Orchard, V.,
Reichstein, M., Rey, A., Ruamps, L., Subke, J. A., Thomsen, I. K. and Chenu, C.: The
moisture response of soil heterotrophic respiration: Interaction with soil properties,
Biogeosciences, 9(3), 1173–1182, doi:10.5194/bg-9-1173-2012, 2012.
Nachtergaele, F.: The classification of Leptosols in the World Reference Base for Soil
Resources, in 19th World Congress of Soil Science, Soil Solutions for a Changing
World, 1–6., 2010.
Packalen, M. S., Finkelstein, S. A. and McLaughlin, J. W.: Climate and peat type in
relation to spatial variation of the peatland carbon mass in the Hudson Bay Lowlands,
Canada, J. Geophys. Res. G Biogeosciences, 121(4), 1104–1117,
doi:10.1002/2015JG002938, 2016.
Parish, F., Sirin, A., Charman, D., Joosten, H., Minayeva, T., Silvius, M. and Stringer,
L.: Assessment on Peatlands, Biodiversity and Climate Change: Main Report., 2008.
Parmentier, F. J. W., van der Molen, M. K., de Jeu, R. A. M., Hendriks, D. M. D. and
Dolman, A. J.: CO2 fluxes and evaporation on a peatland in the Netherlands appear not
affected by water table fluctuations, Agric. For. Meteorol., 149(6–7), 1201–1208,
doi:10.1016/j.agrformet.2008.11.007, 2009.

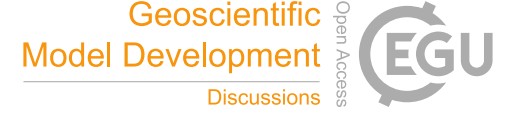

Peltier, W. R.: Global glacial isostasy and the surface of the ice-age Earth: the ICE-5G
(VM2) model and GRACE, Annu. Rev. Earth Planet. Sci., 32, 111–149, 2004.
Portmann, F. T., Siebert, S. and Döll, P.: MIRCA2000-Global monthly irrigated and
rainfed crop areas around the year 2000: A new high-resolution data set for agricultural
and hydrological modeling, Global Biogeochem. Cycles, 24(1),
doi:10.1029/2008GB003435, 2010.
Price, J. S., Cagampan, J. and Kellner, E.: Assessment of peat compressibility: Is there
an easy way?, Hydrol. Process., 19(17), 3469–3475, doi:10.1002/hyp.6068, 2005.
Qiu, C., Zhu, D., Ciais, P., Guenet, B., Krinner, G., Peng, S., Aurela, M., Bernhofer, C.,
Brümmer, C., Bret-Harte, S., Chu, H., Chen, J., Desai, A. R., Dušek, J., Euskirchen, E.
S., Fortuniak, K., Flanagan, L. B., Friborg, T., Grygoruk, M., Gogo, S., Grünwald, T.,
Hansen, B. U., Holl, D., Humphreys, E., Hurkuck, M., Kiely, G., Klatt, J., Kutzbach,
L., Largeron, C., Laggoun-Défarge, F., Lund, M., Lafleur, P. M., Li, X., Mammarella,
I., Merbold, L., Nilsson, M. B., Olejnik, J., Ottosson-Löfvenius, M., Oechel, W.,
Parmentier, F.-J. W., Peichl, M., Pirk, N., Peltola, O., Pawlak, W., Rasse, D., Rinne, J.,
Shaver, G., Schmid, H. P., Sottocornola, M., Steinbrecher, R., Sachs, T., Urbaniak, M.,
Zona, D. and Ziemblinska, K.: ORCHIDEE-PEAT (revision 4596), a model for
northern peatland CO2, water, and energy fluxes on daily to annual scales, Geosci.
Model Dev., 11(2), 497–519, doi:10.5194/gmd-11-497-2018, 2018.
Ringeval, B., Decharme, B., Piao, S. L., Ciais, P., Papa, F., De Noblet-Ducoudré, N.,
Prigent, C., Friedlingstein, P., Gouttevin, I., Koven, C. and Ducharne, A.: Modelling
sub-grid wetland in the ORCHIDEE global land surface model: Evaluation against river
discharges and remotely sensed data, Geosci. Model Dev., 5(4), 941–962,
doi:10.5194/gmd-5-941-2012, 2012.
De Rosnay, P., Bruen, M. and Polcher, J.: Sensitivity of surface fluxes to the number
of layers in the soil model used in GCMs, Geophys. Res. Lett., 27(20), 3329–3332,
doi:10.1029/2000GL011574, 2000.
Roulet, N. T., Lafleur, P. M., Richard, P. J. H., Moore, T. R., Humphreys, E. R. and
Bubier, J.: Contemporary carbon balance and late Holocene carbon accumulation in a
northern peatland, Glob. Chang. Biol., 13(2), 397–411, doi:10.1111/j.1365-
2486.2006.01292.x, 2007.
Sitch, S., Smith, B., Prentice, I. C., Arneth, a., Bondeau, a., Cramer, W., Kaplan, J.
O., Levis, S., Lucht, W., Sykes, M. T., Thonicke, K. and Venevsky, S.: Evaluation of
ecosystem dynamics, plant geography and terrestrial carbon cycling in the LPJ dynamic
global vegetation model , Glob. Chang. Biol., 9(2), 161–185, doi:10.1046/j.1365-
2486.2003.00569.x, 2003.
Smith, L. C.: Siberian Peatlands a Net Carbon Sink and Global Methane Source Since
the Early Holocene, Science, 303(2004), 353–356, doi:10.1126/science.1090553, 2004.
Spahni, R., Joos, F., Stocker, B. D., Steinacher, M. and Yu, Z. C.: Transient simulations
of the carbon and nitrogen dynamics in northern peatlands: From the Last Glacial
Maximum to the 21st century, Clim. Past, 9(3), 1287–1308, doi:10.5194/cp-9-1287-
978    2013, 2013.
Stocker, B. D., Spahni, R. and Joos, F.: DYPTOP: A cost-efficient TOPMODEL
implementation to simulate sub-grid spatio-temporal dynamics of global wetlands and

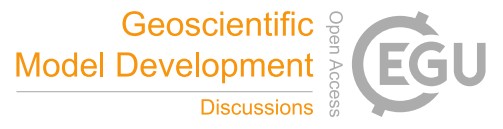

peatlands, Geosci. Model Dev., 7(6), 3089–3110, doi:10.5194/gmd-7-3089-2014, 2014.

Sulman, B. N., Desai, a. R., Cook, B. D., Saliendra, N. and Mackay, D. S.: Contrasting carbon dioxide fluxes between a drying shrub wetland in Northern Wisconsin, USA, and nearby forests, Biogeosciences, 6(6), 1115–1126, doi:10.5194/bg-6-1115-2009, 2009.

Tfaily, M. M., Cooper, W. T., Kostka, J. E., Chanton, P. R., Schadt, C. W., Hanson, P. J., Iversen, C. M. and Chanton, J. P.: Organic matter transformation in the peat column at Marcell Experimental Forest: humification and vertical stratification, J. Geophys. Res. Biogeosciences, 119(4), 661–675, 2014.

Tootchi, A., Jost, A. and Ducharne, A.: Multi-source global wetland maps combining surface water imagery and groundwater constraints, Earth Syst. Sci. Data Discuss., 2018, 1–44, doi:10.5194/essd-2018-87, 2018.

Turetsky, M. R., Amiro, B. D., Bosch, E. and Bhatti, J. S.: Historical burn area in western Canadian peatlands and its relationship to fire weather indices, Global Biogeochem. Cycles, 18(4), 1–9, doi:10.1029/2004GB002222, 2004.

Turetsky, M. R., Benscoter, B., Page, S., Rein, G., Van Der Werf, G. R. and Watts, A.: Global vulnerability of peatlands to fire and carbon loss, Nat. Geosci., 8(1), 11–14, doi:10.1038/ngeo2325, 2015.

Turunen, J., Tahvanainen, T., Tolonen, K. and Pitk??nen, A.: Carbon accumulation in West Siberian mires, Russia, Global Biogeochem. Cycles, 15(2), 285–296, doi:10.1029/2000GB001312, 2001.

Turunen, J., Tomppo, E., Tolonen, K. and Reinikainen, A.: Estimating carbon accumulation rates of undrained mires in Finland – application to boreal and subarctic regions, The Holocene, 12(1), 69–80, doi:10.1191/0959683602hl522rp, 2002.

Wania, R., Ross, I. and Prentice, I. C.: Integrating peatlands and permafrost into a dynamic global vegetation model: 1. Evaluation and sensitivity of physical land surface processes, Global Biogeochem. Cycles, 23(3), 1–19, doi:10.1029/2008GB003412, 2009a.

Wania, R., Ross, I. and Prentice, I. C.: Integrating peatlands and permafrost into a dynamic global vegetation model: 2. Evaluation and sensitivity of vegetation and carbon cycle processes, Global Biogeochem. Cycles, 23(3), 1–15, doi:10.1029/2008GB003413, 2009b.

Wu, Y., Verseghy, D. L. and Melton, J. R.: Integrating peatlands into the coupled Canadian Land Surface Scheme (CLASS) v3.6 and the Canadian Terrestrial Ecosystem Model (CTEM) v2.0, Geosci. Model Dev., 9(8), 2639–2663, doi:10.5194/gmd-9-2639-2016, 2016.

Wu, Y., Chan, E., Melton, J. R. and Verseghy, D. L.: A map of global peatland distribution created using machine learning for use in terrestrial ecosystem and earth system models, Geosci. Model Dev. Discuss., (July), 2017.

Xu, J., Morris, P. J., Liu, J. and Holden, J.: PEATMAP: Refining estimates of global peatland distribution based on a meta-analysis, Catena, 160(September 2017), 134–140, doi:10.1016/j.catena.2017.09.010, 2018.

Yu, Z.: Holocene carbon flux histories of the world's peatlands: Global carbon-cycle implications, The Holocene, 21(5), 761–774, doi:10.1177/0959683610386982, 2011.



Yu, Z., Campbell, I. D., Campbell, C., Vitt, D. H., Bond, G. C. and Apps, M. J.: Carbon sequestration in western Canadian peat highly sensitive to Holocene wet-dry climate cycles at millennial timescales, The Holocene, 13(6), 801–808, doi:10.1191/0959683603hl667ft, 2003a.

Yu, Z., Vitt, D. H., Campbell, I. D. and Apps, M. J.: Understanding Holocene peat accumulation pattern of continental fens in western Canada, Can. J. Bot., 81(3), 267–282, 2003b.

Yu, Z., Beilman, D. W. and Jones, M. C.: Sensitivity of Northern Peatland Carbon Dynamics to Holocene Climate Change, Carbon Cycl. North. Peatlands, 184, 55–69, doi:10.1029/2008GM000822, 2009.

Yu, Z., Loisel, J., Brosseau, D. P., Beilman, D. W. and Hunt, S. J.: Global peatland dynamics since the Last Glacial Maximum, Geophys. Res. Lett., 37(13), 1–5, doi:10.1029/2010GL043584, 2010.

Yu, Z. C.: Northern peatland carbon stocks and dynamics: A review, Biogeosciences, 9(10), 4071–4085, doi:10.5194/bg-9-4071-2012, 2012.

Zaccone, C., Sanei, H., Outridge, P. M. and Miano, T. M.: Studying the humification degree and evolution of peat down a Holocene bog profile (Inuvik, NW Canada): A petrological and chemical perspective, Org. Geochem., 42(4), 399–408, doi:10.1016/j.orggeochem.2011.02.004, 2011.

Zhang, Z., Zimmermann, N. E., Kaplan, J. O. and Poulter, B.: Modeling spatiotemporal dynamics of global wetlands: Comprehensive evaluation of a new sub-grid TOPMODEL parameterization and uncertainties, Biogeosciences, 13(5), 1387–1408, doi:10.5194/bg-13-1387-2016, 2016.

Zhu, D., Peng, S. S., Ciais, P., Viovy, N., Druel, A., Kageyama, M., Krinner, G., Peylin, P., Ottlé, C., Piao, S. L., Poulter, B., Schepaschenko, D. and Shvidenko, A.: Improving the dynamics of Northern Hemisphere high-latitude vegetation in the ORCHIDEE ecosystem model, Geosci. Model Dev., 8(7), 2263–2283, doi:10.5194/gmd-8-2263-2015, 2015.







**Table 1.** Measured and simulated minimum, maximum and median depth (m) of peat
cores, grouped by peatland types, ages, and climatic regions. The root mean square
errors between observations and simulations are also listed.

|  | Measured | | | Simulated | | | |
|---|---|---|---|---|---|---|---|
|  | Minimum | Maximum | Median | Minimum | Maximum | Median | RMSE |
| Fens | 1.10 | 7.25 | 3.78 | 0.75 | 4.30 | 2.16 | 2.08 |
| Bogs | 0.96 | 10.95 | 3.30 | 0.75 | 5.49 | 2.18 | 2.59 |
| Others | 1.00 | 3.95 | 1.94 | 0.37 | 6.64 | 2.38 | 2.46 |
| 12 ka ≤ Age | 2.45 | 8.61 | 3.52 | 0.37 | 3.21 | 2.64 | 2.78 |
| 10 ≤ Age < 12 ka | 1.28 | 7.24 | 3.60 | 1.50 | 5.40 | 3.20 | 2.72 |
| 8 ≤ Age < 10 ka | 1.89 | 10.95 | 3.25 | 0.75 | 6.64 | 2.16 | 3.33 |
| 6 ≤ Age < 8 ka | 0.96 | 4.82 | 3.00 | 0.75 | 5.49 | 2.15 | 1.54 |
| 4 ≤ Age < 6 ka | 1.00 | 5.75 | 2.44 | 0.75 | 2.18 | 1.54 | 1.73 |
| Arctic | 1.00 | 5.10 | 1.80 | 0.97 | 5.48 | 3.39 | 2.25 |
| Boreal | 0.96 | 10.95 | 3.22 | 0.37 | 6.64 | 2.15 | 2.35 |
| Temperate | 3.09 | 7.24 | 6.17 | 1.50 | 3.20 | 2.18 | 3.98 |
| All | 0.96 | 10.95 | 3.10 | 0.37 | 6.64 | 2.18 | 2.45 |















**Table 2.** Observed (estimates from peatland inventories and soil database) and
simulated northern peatland area, countries are sorted in descending order according to
the estimate of IMCG-GPD.

| country/area | Peatland area ($10^3$ km$^2$) | | | |
|---|---|---|---|---|
| | IMCG-GPD | WISE | PEATMAP | Simulated $f_{\text{noLEP-CR}}$ |
| >30°N | >3000 | 2823 | 3250 | 3896 |
| Russia-Asian part | 1176 | 852 | 1217 | 1336 |
| Canada | 1134 | 1031 | 1095 | 1009 |
| Russia-European part | 199 | 285 | 207 | 392 |
| USA(Alaska) | 132 | 167 | 72 | 168 |
| USA(lower 48) | 92 | 49 | 98 | 365 |
| Finland | 79 | 89 | 69 | 42 |
| Sweden | 66 | 65 | 58 | 35 |
| Norway | 30 | 19 | 14 | 29 |
| Mongolia | 26 | 13 | 13 | 6 |
| Belarus | 22 | 29 | 22 | 11 |
| United Kingdom | 17 | 21 | 17 | 42 |
| Germany | 17 | 14 | 13 | 33 |
| Poland | 12 | 18 | 16 | 8 |
| Ireland | 11 | 9 | 14 | 17 |














**Table 3.** Observed and simulated northern peatland C, countries are sorted
in descending order according to the estimate of IMCG-GPD.

| country/area | Peat carbon stock (Pg C) | | |
|---|---|---|---|
| | IMCG-GPD | WISE | Simulated $f_{noLEP-CR}$ |
| >30°N | | 421 | 463 |
| Canada | 155 | 155 | 87 |
| Russia-Asian part | 118 | 114 | 174 |
| Russia-European part | 20 | 38 | 49 |
| USA(Alaska) | 16 | 28 | 32 |
| USA(lower 48) | 14 | 10 | 45 |
| Finland | 5 | 15 | 5 |
| Sweden | 5 | 10 | 4 |
| Norway | 2 | 3 | 3 |
| Germany | 2 | 3 | 5 |
| United Kingdom | 2 | 4 | 7 |
| Belarus | 1 | 4 | 1 |
| Ireland | 1 | 2 | 4 |








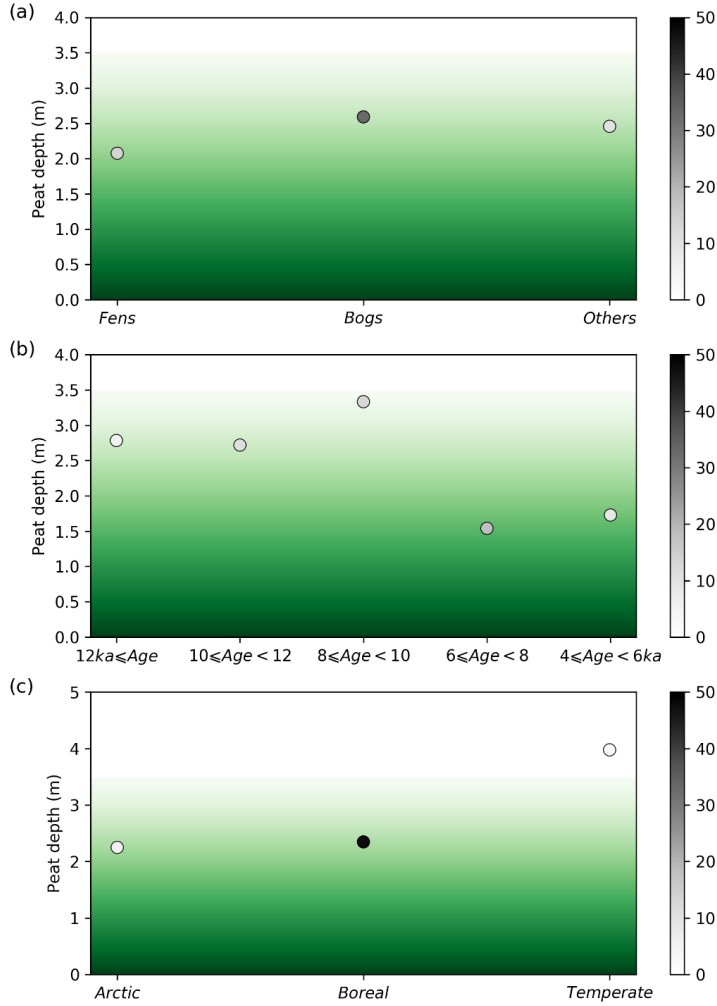



**Fig. 1.** Root mean square error (RMSE) of measured and simulated peat depth at 60
peatlands sites (Table S1), grouped by peatland types (a), ages (b), and climatic regions
(c). The transition from green to white indicates an RMSE of 100 %. Number of sites
included in the calculation is showed by colors of the symbols.







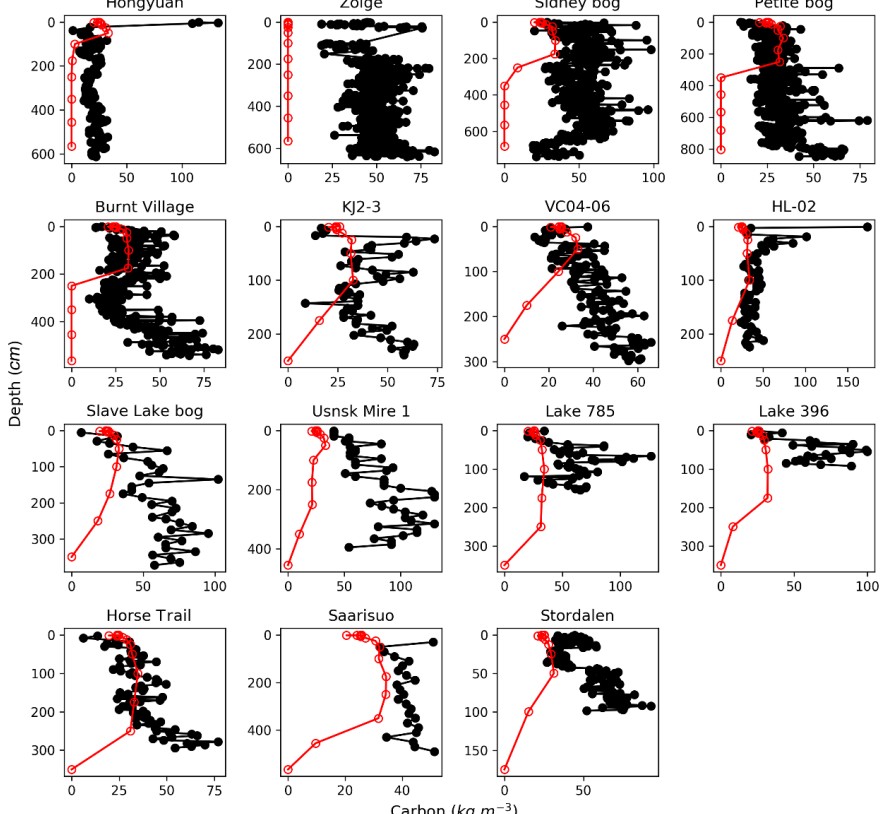

**Fig. 2.** Observed (black) and simulated (red) vertical profiles of soil C, at the 15 sites
where peat age, depth, bulk density and carbon fraction have been measured (Table S1).
The black circles indicate depths of measurements, the red circles indicate the depth of
each soil layer in the model.






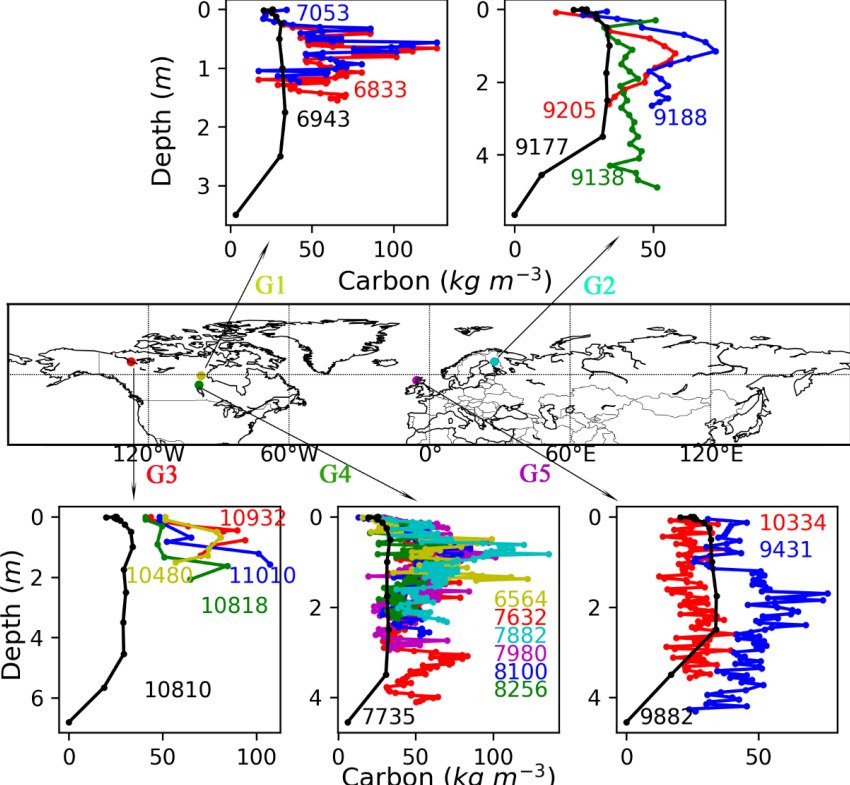


**Fig. 3.** Observed (colored, with each colored line represent one peat core) and simulated
(black) vertical C profiles of five grid cells where there is more than one core. The
numbers in the figure indicate ages of sampled peat cores (colored) and time length of
the simulation (black, is the mean age of all cores in the same grid cell).

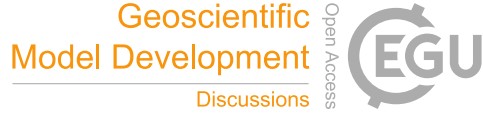




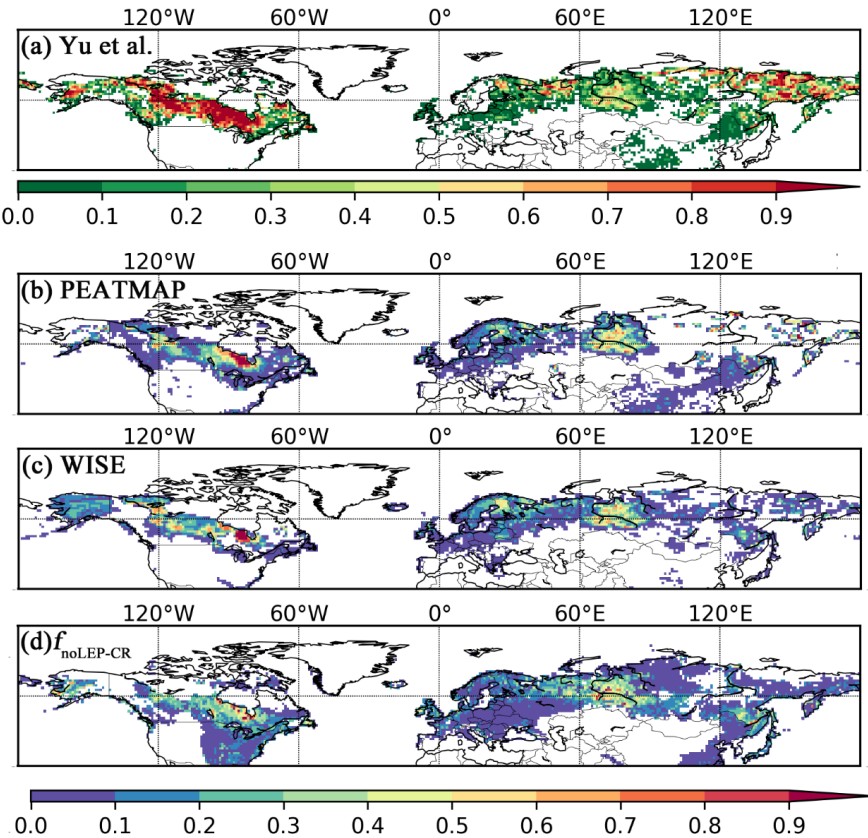



**Fig. 4.** Observed and simulated peatland area fraction. (a) Peatland fractions obtained from qualitative map of Yu et al. (2010). The original qualitative map only delineates areas with peatland coverage greater than 5%, the quantitively data here is derived by aggregating the interpolated $0.05° \times 0.05°$ grid cells into $1° \times 1°$ fractions, thus it's not directly comparable to the fractional peatland area of other datasets and the model output. We illustrate it with a distinct color key, (b) peatland area fraction derived from the PEATMAP, (c) histosol fractions from the WISE soil database, (d) simulated peatland area fraction ($f_{noLEP-CR}$), with pattern and timing of deglaciation has been considered. Areas dominated by Leptosols has been masked and areas occupied by crops has been excluded, under the assumption that cropland occupied peatland in proportion to grid cell peat fraction.






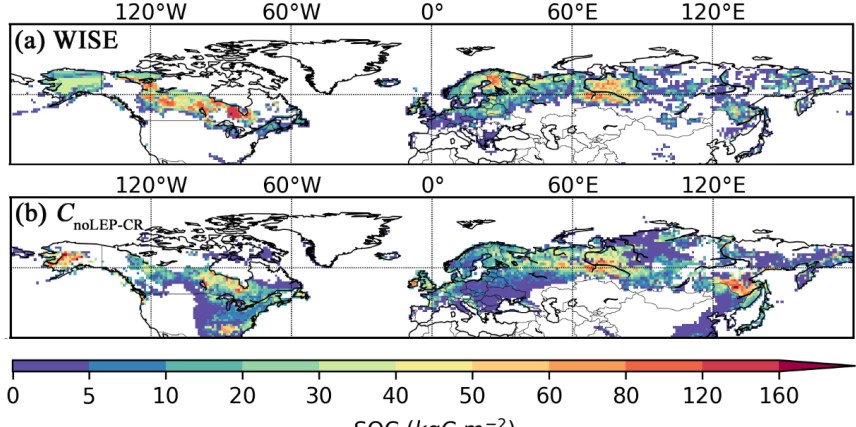


**Fig. 5.** Observed and simulated peatland soil carbon density. (a) Peatland (Histosols)
soil carbon density from the WISE soil database, (b) simulated peatland soil carbon
density ($C_{\text{noLEP-CR}}$), with pattern and timing of deglaciation has been considered. Areas
dominated by Leptosols has been masked and areas occupied by crops has been
excluded, under the assumption that cropland occupied peatland in proportion to grid
cell peat fraction.



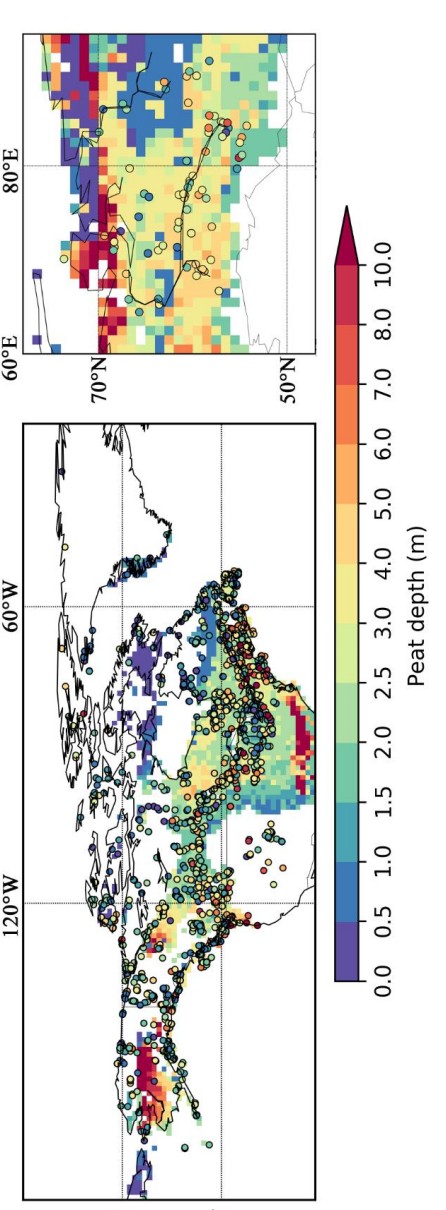

**Fig. 6.** Measured (color filled circles, with colors indicating measured values) and simulated (background maps) peat depth in North America (left) and in the West Siberian lowlands (right). Measured peat cores from North America are from Gorham et al. (2012), while that from the West Siberian lowlands are from Kremenetski et al. (2003).






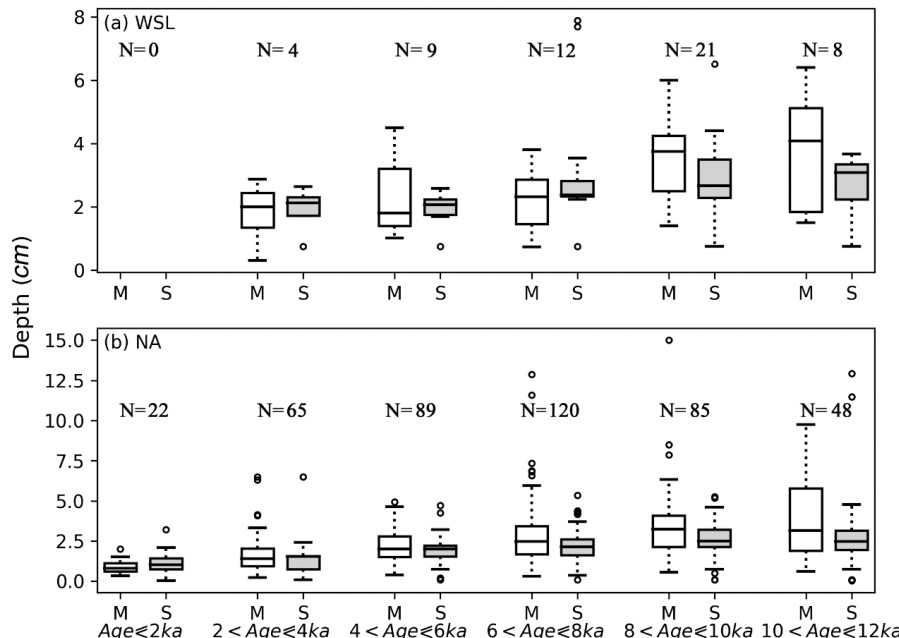


**Fig. 7.** Measured (M) and simulated (S) mean peat depth at the West Siberian lowlands
(a) and North America (b), grouped according to the mean age of peat cores. Measured
peat cores are from Gorham et al. (2012) and Kremenetski et al. (2003). The horizontal
box lines: the upper line - the 75th percentile, the central line - the median (50th
percentile), the lower line - the 25th percentile. The dashed lines represent 1.5 times the
IQR. The circles are outliers. Number of included grid cells in each age group is
indicated by N.







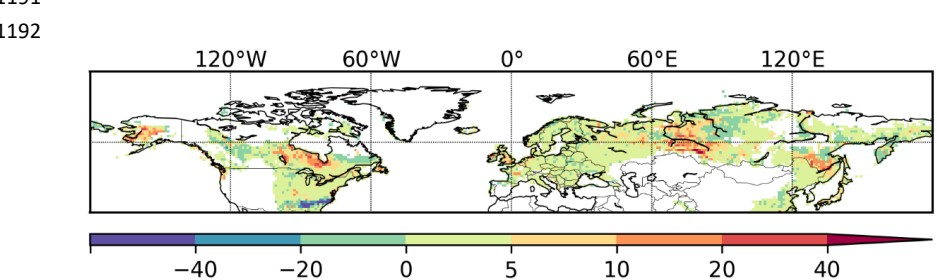


**Fig. 8.** Simulated annual net ecosystem production (NEP), averaged over 1901 – 2009.
Obtained by multiplying peatland NEP (gC m$^{-2}$ peatland a$^{-1}$) with peatland fraction for
each grid cell.








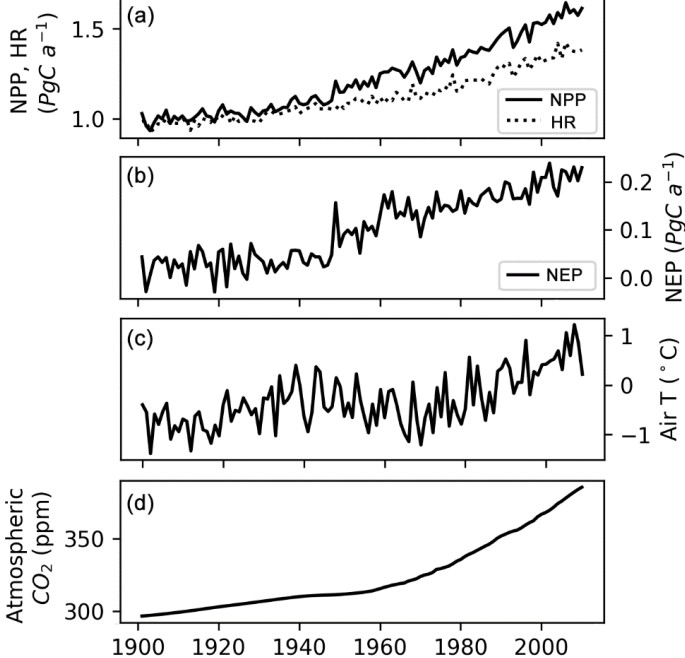



**Fig. 9.** (a) Simulated annual net primary production (NPP), heterotrophic respiration
(HR) of northern peatlands, (b) simulated net ecosystem production (NEP) of northern
peatlands, (c) mean air temperature (T) of grid cells that have peatland, (d) atmospheric
$CO_2$ concentration.









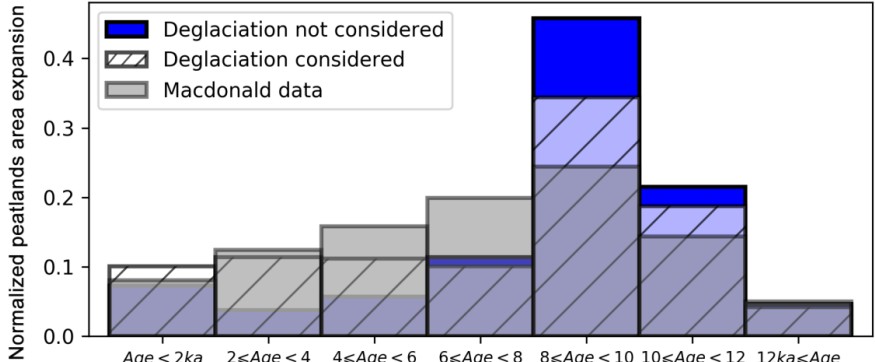


**Fig. 10.** (Grey bars) Percentage of observed peatland initiation (grey) in 2000-year bins.
Peat basal dates of 1516 cores are from MacDonald et al. (2006), peat basal age
frequency of each 2000-year bin is divided by the total peat basal age frequency. (Blue
bars) Percentage of simulated peatlands area developed in each 2000-year bin,
deglaciation of ice-sheets is not considered (the model was run with 6 times SubC, 2000
years each time). The peatlands area developed in each bin is divided by the simulated
modern (the year 2009) peatlands area. (White hatched bars) Percentage of simulated
peatlands area developed in each 2000-years bin, pattern and timing of deglaciation are
read from maps in Fig. S5 and Fig. S6.