# Peer review of "Modelling northern peatlands area and carbon dynamics since the Holocene with"

_Geoscientific Model Development, 2018_

## Referee Comment (RC1) · Stocker (Referee) · 11 Jan 2019

This paper presents and evaluates a global model that simulates the spatial extent of peatlands and their C balance as a function of the environment. The peatland model is implemented as a module within the comprehensive land surface model ORCHIDEE. This is an important addition to this model as it allows to account for the effect of peatlands on the global carbon cycle, which is particularly important for long-term simulations, covering multiple centuries to the millennial time scale. The approach for simulating the spatial dynamics of peatlands across the globe is largely adopted from Stocker et al. (2014) GMD [thereafter referred to as ST14]. I don't want to hide the

fact that this is my own work and that I am pleased to see that it has stimulated other researchers to follow the same approaoch.

The paper by Qiu et al. goes a step beyond ST14 in that it evaluates the model not only by its accurateness in simulating the spatial patterns across the globe and the total northern peatland C storage, but it evaluates peat depth using information from a set of 102 peat cores, distributed across the northern hemisphere (mostly in the boreal zone), and deals with the challenge of accurately simulating the history of peat C accumulation throughout the Holocene, which adds substantial complexity. This work is also a substantial advancement in simulating wetlands and the distribution of flooding. Their comparison to a new observation-based dataset by Tootchi et al. (2018) shows a very good agreement (Fig. S7 - worth including this in the main text?), and seems to suggest that their model works much better in this respect than, e.g., the model presented in ST14. This in itself is a very useful innovation. I was also intrigued by the clever approach to simulate vertical growth of peat as an effective downward transport of soil C (down along the soil profile, across the 32 layers resolved by the model). This is a very useful innovation beyond the models resolved by ST14 and Kleinen et al., 2012. I think this work can be a very valuable addition to the literature and that the model presented here will be a useful addition to the very small set of comparable models available today (only two models, as I am aware). However, before getting there, I would like to see a few critical (MAJOR) issues addressed. I also think that the paper could gain from a clearer presentation in general. Below, I'm listing specific points. I hope the authors find my suggestions useful and I am looking forward to a revised version of the manuscript, and possibly a revision of the model and evaluation.

MAJOR

* The code is not accessible under the given URL. Although it's not offcially required by GMD, I personally try to resist to accept model description papers without having open access code. I also think that the model should be easily reproducible in a simplified setup (without having to run the entire ORCHIDEE) and instructions should be

available to do so. Plug and play! Please make an effort to achieve this, it is greatly appreciated by the community and helps science to move forward (and it pays off for you).

* What the paper/model does not tackle/resolve, goes unmentioned. No tropical peatlands are simulated (?) nor evaluated. Are methane emissions from peatlands not resolved by the model? How does peat vs. mineral soil affect the extent of frozen soils (permefrost!)? The evaluation of inundation, particularly its timing is missing (or hidden in the SI).

* The simulated distribution of the peatland area fraction (Fig. 4) shows that the model is able to broadly capture the observed pattern, except that is quite strongly underestimates the peatland extent in the Hudson Bay Lowland (HBL). This reminds me of my own work, where the first version of my model (DYPTOP, ST14) also failed to simulate very high peatland area fractions (over 90%) across this large region. The HBL is, next to the West Siberian Lowland, the largest peatland region and therefore warrants special attention. The failure of the model by Qiu et al. in simulating large peatland fractions may be related to what one may call the "sponge-feedback" - the high efficiency of organic soils in retaining water (small runoff) which in turn increases persistency of flooding and the suitability for peat to accumulate - a positive feedback. I solved this by having (gridcell average) soil parameters that determine the soil hydrology depending on the internally simulated peatland area fraction, rather than using externally prescribed parameters from soil maps. I see that in the present model, some soil parameters are indeed prescribed for each gridcell separately from external data (soil bulk density, soil C fraction; l. 499). I would say that they should be affected by whether the model simulates peatland in the respective gridcell or not. This might be something worth looking into in order to better reproduce the observed Hudson Bay Lowland peatland area fractions. On l.131, it's mentioned that soil thermal (and hydrological?) properties are a weighted average of mineral and organic soils, where organic soil fraction is prescribed from an external dataset (NCSCD and HWSD).

* The explicit depth-dependence of the turnover rates is a bit obscure to me. While the rationale is defensible (l. 160 " priming effects, sorption of organic molecules to mineral surfaces"), it's not clear how important this factor is for the simulations here. Couldn't it be avoided? What's the e-folding scale in Eq. 2? (I see that the $z_0$ parameter is given later in the manuscript) And shouldn't this be accounted for by oxygen conditions, being subject to water content in different layers where the bottom layers will tend to be water-logged and thus have a very low turnover rate. From text S3, this is not evident.

* Comparison with cores. I am not sure if the model presented here can be compared to peat cores. The reason is that, in order to conserve C mass, an expansion of the peatland area fraction has to imply a reduction of the peat C mass per unit area - peat C is effectively diluted over an increasing area. Hence, the vertical growth of peat should slow upon lateral expansion. This is implied by the simplification that the model doesn't explicitly simulate the horizontal dimension. In reality, a peatland has substantial lateral structure and tends to be deep and have the oldest layers towards the center. That's also where peat cores are commonly taken (in order to maximise the temporal coverage). I am therefore not surprised to see that the model appears to generally underestimate peat depth. I suspect that separate simulations are required for this, where the peatland area fraction is held constant (no dilution!).

* The authors aim to model peat C dynamics during the Holocene (see title), but relatively little focus is given to forcing and evaluating the model with respect to this palaeo perspective. As far I understood, the model is forced with constant pre-industrial climate (although insolation and summer temperatures varied substantially during the Holocene, especially at high latitudes). Was a changing sea level accounted for? For applications in palaeo climate and -carbon cycle studies, the model is expected to reliably simulate the net C balance of peatlands. I am not convinced that the evaluation of C content across the soil profile, as presented in the paper, provides sufficient information to evaluate this aspect. Shouldn't a comparison be done against dated peat cores, where the amount of C (left today) per age bin is given? The model doesn't track

age bins explicitly, but could be extended to simulate C14 decay and transport across the soil layers (so that lower layers would have an older C14 age, which could then be compared to the C14 age across depth in dated cores). Alternatively, one could write out soil C inputs and decomposition rates at all time steps and resolve age cohorts explicitly offline (diagnostically). I understand that this is a substantial challenge, but I am not fully convinced that the evaluation presented here is sufficient. At least a discussion of these points should be added.

* I simply did not understand Fig. 1.

* Should become clear upfront what parameters are calibrated and what observational targets are used for calibration.

LESS MAJOR (BUT NOT MINOR)

* Better define the scope of the model and the evaluation, the scale at which the model is expected to yield reliable results, what simplifications have taken to get there, and where the model is not applicable. This can be achieved by more clearly stating upfront for what research questions the model is expected to be applied, and what it therefore needs to simulate with fidelity (and why these quantities). And then present the results with a focus and structure to address these quantities. This is largely done so already, but it would greatly help the reader to improve the structure of the paper in this sense. I would expect the following key quantities:

* total (northern) peat C: ok

* spatial patterns of peatland extent: ok, although the particularly extensive peat area in the Hudson Bay Lowland is largely missed by the model.

* basal age/inception, compared to first year of peatland establishment in model: It would be good to evaluate simulated and observed basal ages across space, e.g. with a map showing the simulated dbasal age across space and dots on top of it for observed basal ages from different cores.

* peat C accumulation/respiration history: The net C balance through time is what is relevant for the C cycle (what the atmosphere "sees"). I am not convinced that the evaluation presented here, looking at C content across depth, is giving us the right information to evaluate the model in this respect. The dimension time is missing (as mentioned above); there is no age scale of the cores factored into the analysis.

* Vertical peat growth model: I didn't intuitively understand the rationale for using bulk density data to formulate the vertical growth/downward transport model. Why didn't you use volumetric C content? Can your approach be described as a sequence of C-buckets that fill up by receiving inputs from the layer above (once this "spills over")? Then, spill-over is happening when the typical empirical volumetric C density at the respective depth, as measured in your 102 cores, is achieved. I'm just thinking out loud here, trying to make sense of the model. But maybe you can include such an intuitive description of your approach in the paper.

* While the striking performance in simulating inundation is definitely a plus, it remains unclear how this improvement over earlier publications (e.g., ST14) is achieved. Is it related to resolving the soil hydrology across layers instead of using a simple bucket model? The inundation sub-model is key for the peatland extent model and warrants a bit more attention in the paper.

* I don't think it's appropriate to require every model presented in GMD to be fundamentally novel. Furthermore, the model presented by Qiu et al. is largely an adoption of an existing model (ST14), which itself is based on Kleinen et al., 2012. Sufficient reference is made by Qiu et al. to this earlier work. However, the authors introduce and motivate their work with (l.94) "While both studies made pioneering progresses in the modelling of peatland ecosystems, they adopted a simple bucket approach to model peatland hydrology and peatland C accumulation, and neither of them resolved the diel cycle of surface energy budget." However, it is unclear why the diurnal surface energy budget needs to be explicitly simulated in this context, and what limitations the simple bucket model approach incurs. It definitely needs more clarification what the model

adds to our knowledge and our predictive power and I am sceptical that resolving the diurnal cycle of surface energy exchange adds a great deal. I am more curious about whether resolving soil hydrology across multiple layers helps better simulating relevant peatland-related processes, but the paper doesn't provide this insight. I think it is important that it becomes better clear what the merit of this model (over existing ones) is.

* Observed (Mc Donald et al., 2006) and modelled inception age could be compared across space rather than just showing the numbers across time in Fig. 10. Actually, this comparison is subject to a possible sampling bias in Mc Donald. You want to test whether the model simulates the right inception time at a specific location, and not only the fraction of total number of simulated against the total number of sampled peatlands sampled in each age bin.

MINOR

* l.21: I wouldn't subscribe to 'recently'.

* l.34: "270-540 PgC" Seems to be at the low end. What's the reference? On l. 44 references are given. But I suggest to use the latest (Yu, 2010) as the benchmark.

* l.48 "in environments..." Make a new sentence, as this is not related to the first part of the sentence

* l.49: Change 'despite' to 'while'.

* l.64/65: Weird sentence. The depth itself doesn't prevent oxygen supply.

* l. 69: Unclear: "critical level [of WTD???]"

* l.69-74: This sounds like the authors highlight a unresolved challenged here that the model/paper is going to address. However, it's unclear what is meant here (of course, WTD determines soil moisture or vice-versa), and how the model and results presented here address this particular challenge.

* l. 70: Isn't WTD linearly related to soil moisture content? Why the threshold?

* l.76: Style: don't refer to 'groups'.

* l. 92: I would say that the key in ST14 was to account for peatland-specific water storage capacity in typical organic soils ("sponge" feedback) which enabled to accurately simulate the particular patter of peatland areas across the globe.

* l. 98: Unclear what "discrepancies" are referred to.

* l.121: 'multi' instead of 'many'

* Eq. 4: Why isn't it flux = f * (C_l - M_th,l)? The way it's formulated, the C content may drop below the threshold after transfer. Shouldn't it stay "saturated" after accounting for downward transport?

* Eq. 5: What's the rationale for introducing parameter f_th? Why isn't it 1?

* l. 216: . . . than what? Explanation would be helpful: More computationally efficient than determining water table depth for each sub-grid pixel.

* l. 227: Ambiguous formulation. Do you mean max(monthly values)?

* Section 2.2.2.: Put Fig. S2 into main text and highlight difference to ST14.

* l. 238: Not quite correct. I don't know what the authors refer to here.

* l. 249: Question: Are non-growing season months discarded or do they count towards N?

* l. 254: What's the difference to ST14? Using only water balance during summer months instead of entire year? Might be worth mentioning explicitly. What's the rationale for this choice? Note that winter precipitation is relevant too as summer snow melt is effectively delayed winter precipitation.

* l. 258 "May-September": Warning: this would mean that the model is not applicable in the south.

[Figure]

* l. 270: Clarify: C_lim "is defined here as. . ."

* l. 278: "SubC": What's this?

* l. 280-283: Add reference to ST14 as this is the same procedure as chosen by them.

* l. 350: Missing references

* l. 371: Figure for modelled vs. observed depth would be instructive.

* Fig 1/Sect. 4.1: I didn't understand Fig. 1 and how I can read the RMSE from that figure. I expected a comparison of modelled and observed peat depth (or total column C), possibly split by temperate/boreal/arctic and/or bog/fen.

* l. 411: Worth including figure in main text.

* l. 425: Are leptosols and agricultural peatlands simply deducted from simulated peatland areas?

* l. 440: Abbreviations introduced?

* l. 455: "we can only make the..." I don't understand this part.

* l. 489: "several": delete

Beni Stocker

---

## Referee Comment (RC2) · Anonymous Referee #2 · 1 Feb 2019

Qiu et al. present their new peatland model, ORCHIDEE-PEAT (v2) and use it prognostically simulate peatland C, extent, and depth over the Holocene. Their work borrows from previous efforts using TOPMODEL based approaches but they extent the field by allowing their model to determine where peatlands will initiate and expand. I find the work to be on the whole sound and interesting. The problem they are tackling is far from trivial and I am surprised it does as well as it does. I am a little concerned about the poorer performance in the major peatland complexes of the world (Hudson's Bay and West Siberia) which I get to in my comments. The paper is generally easy to follow and has relatively few typographical/grammatical errors. I think the paper is publishable in GMD but would like to see my comments addressed prior to that.

[Figure]

Main comments:

1. The paper seems to sometimes confuse wetlands and peatlands. While peatlands are a type of wetland, in the paper the distinction can be at times very fuzzy. For example, in the abstract it says 'A cost-efficient TOPMODEL approach is implemented to simulate the dynamics of peatland area, calibrated by present-day wetlands areas that are regularly inundated or subject to shallow water tables' (lines 28 - 30). Since it is very possible to have a non-peatland wetland be 'regularly inundated or subject to shallow water tables' this makes it confusing at a minimum. Later in the supplementary material some model parameters are tuned, grid cell by grid cell, to 'select the combination that matches with the CW-WTD wetlands map'. So it appears quite unclear that this is indeed a peatland specific parameterization. I realize that there are other steps to determine if peat will begin to form at the site (e.g. Fig S2) but the implementation of the wetland/peatland determination scheme is confusing. Why tuned to wetland area if that will include many non-peat wetlands? Is the idea that the peatland initiation scheme can handle the rest? Can the authors try and bring a bit more clarity to that aspect of their technique?

2. I fully understand the authors' point about difficulty in simulating small permafrost complexes (e.g. discussion of Fig 6) but I am concerned about the poorer performance in the major complexes such as the HBL or WSL. Both of these regions have areas of near 100% peatland cover so the model should have a good chance. Also there is an overabundance of peatlands in some regions that are generally devoid of peatlands (e.g. E. USA). Is this 'smearing' of peatlands perhaps a result of how wetlands area is generally determined, i.e. TOPMODEL-based, or is this a result of the peat initiation limits? I think this deserves more discussion in the paper as it is a striking aspect of the result and one that the community would benefit from any lessons learned regarding how to best get the hotspots without overdoing the rest of the domain.

Minor comments:

1. line 202 - does that mean the peatland PFTs are forced into their gridcells? Can you expand on what peatland PFTs there are? I see that there are some mention in Text S1 but it just says a PFT with shallow roots. Is it a tree? Do you simulate any other peatland specific PFTs? Shrubs? Moss? Sedges?

2. line 224 - Since Fan et al. 2013 is a model-based product perhaps add in 'simulated' in the description.

3. line 265 - Does the peatland HSU immediately shrink to the new potential peatland area fraction? No lag or delay?

4. line 282 - Why is the old peat treated as mineral soils? That strikes me as strange. The soils would continue to have high C contents for quite a while if drained so treating as a mineral soil seems unreasonable. Please expand on this logic.

5. line 400 - Didn't understand the last sentence there.

6. line 447 - How many cores were simulated as non-peat out of the total?

7. around line 476 - please specify 'simulated'. It gets a bit confusing that these are all just model quantities.

8. line 626 - This is where I find the technique a bit confusing. 'We notice a large interannual variability in peatland area'. In reality this is unlikely to be possible given that peat soils are slow to develop and slow to leave. The water-logging is the dynamic aspect. This sort of ties into my main comment #1 above. Please tighten up how this is all defined and referred to.

9. Fig 1 - Strange figure. I couldn't figure out the green fade, nor understand how it was giving information. So is the above the green the >100% RMSE? Why a fade? Please rethink this one.

10. If Fig 6 is plotted as a simple scatterplot, what does it look like? I understand that Fig 7 is a more detailed look but I wonder if a simple scatter plot could be instructive

for any bias.

11. Fig 10 - please split into 3 separate bars per time period. I couldn't figure this out. What is the light blue? What is the line midway through 8-10 Age bar meaning?

12. supplementary line 11 - So does all of the surface runoff from the grid cell get funnelled into the peatland HSU? Why only surface and not subsurface?

p.s. Apologies for the slow review. There was some confusion between me and the editorial team if I was providing a review.

---

## Author Comment (AC1) · 28 Mar 2019

We would like to thank Benjamin Stocker and the anonymous referee very much for their constructive comments. In the following, please find our point by point response to the comments.

- Reviewer's comments are in bold
- Modifications done in the revised manuscript are in blue
- All figure numbers, table numbers, and line numbers refer to the initial manuscript version.

**Stocker (Referee)**

**This paper presents and evaluates a global model that simulates the spatial extent of peatlands and their C balance as a function of the environment. The peatland model is implemented as a module within the comprehensive land surface model ORCHIDEE. This is an important addition to this model as it allows to account for the effect of peatlands on the global carbon cycle, which is particularly important for long-term simulations, covering multiple centuries to the millennial time scale. The approach for simulating the spatial dynamics of peatlands across the globe is largely adopted from Stocker et al. (2014) GMD [thereafter referred to as ST14]. I don't want to hide the fact that this is my own work and that I am pleased to see that it has stimulated other researchers to follow the same approach.**

**The paper by Qiu et al. goes a step beyond ST14 in that it evaluates the model not only by its accurateness in simulating the spatial patterns across the globe and the total northern peatland C storage, but it evaluates peat depth using information from a set of 102 peat cores, distributed across the northern hemisphere (mostly in the boreal zone), and deals with the challenge of accurately simulating the history of peat C accumulation throughout the Holocene, which adds substantial complexity. This work is also a substantial advancement in simulating wetlands and the distribution of flooding. Their comparison to a new observation-based dataset by Tootchi et al. (2018) shows a very good agreement (Fig. S7 - worth including this in the main text?), and seems to suggest that their model works much better in this respect than, e.g., the model presented in ST14. This in itself is a very useful innovation. I was also intrigued by the clever approach to simulate vertical growth of peat as an effective downward transport of soil C (down along the soil profile, across the 32 layers resolved by the model). This is a very useful innovation beyond the models resolved by ST14 and Kleinen et al., 2012. I think this work can be a very valuable addition to the literature and that the model presented here will be a useful addition to the very small set of comparable models available today (only two models, as I am aware). However, before getting there, I would like to see a few critical (MAJOR) issues addressed. I also think that the paper could gain from a clearer presentation in general. Below, I'm listing specific points. I hope the authors find my suggestions useful and I am looking forward to a revised**

**version of the manuscript, and possibly a revision of the model and evaluation.**

We thank the reviewer for his thorough reading of this manuscript and encouraging comments. We include Fig. S7 in the main text of the revised manuscript. Please see below our detailed response to comments.

**MAJOR**

**Q1. * The code is not accessible under the given URL. Although it's not officially required by GMD, I personally try to resist to accept model description papers without having open access code. I also think that the model should be easily reproducible in a simplified setup (without having to run the entire ORCHIDEE) and instructions should be available to do so. Plug and play! Please make an effort to achieve this, it is greatly appreciated by the community and helps science to move forward (and it pays off for you).**

The source code is freely available and accessible via the following address: https://forge.ipsl.jussieu.fr/orchidee/wiki/GroupActivities/CodeAvalaibilityPublication/ORCHIDEE_PEAT_V2

Moreover, we agree with the reviewer that a simplified version of the model using some kind of emulator will be helpful for interested readers. However, the ORCHIDEE-PEAT model simulates both carbon and area dynamics of peatland, which consists of the following hydrological and biogeochemical processes and their interactions (non-exhaustive): 1. Physically-based soil water flows and soil moisture constrain area development of peatland. Meanwhile, peatland receives water input from surrounding mineral soils, increases soil water storage and reduces runoff of the grid cell, thus exerts a feedback effect on soil water dynamics; 2. Soil moisture limits phenology, photosynthesis, transpiration and soil thermics, which in turn impact the water cycle; 3. Soil hydrology and soil thermics impact litter and soil carbon decomposition, while the long-term C balance of the peatland limits peatland area development. All those mechanisms feedbacks on each other and the design of an emulator will be a research project as itself.

**Q2. * What the paper/model does not tackle/resolve, goes unmentioned. No tropical peatlands are simulated (?) nor evaluated. Are methane emissions from peatlands not resolved by the model? How does peat vs. mineral soil affect the extent of frozen soils (permafrost!)? The evaluation of inundation, particularly its timing is missing (or hidden in the SI).**

We didn't simulate tropical peatlands in this study, because the model is parameterized and calibrated for northern peatlands. To clarify this point, we add sentences on Line657: "Being parameterized and calibrated for northern peatlands, our model can't be used for tropical peatlands. For tropical peatlands, the model needs to be improved to represent its tree dominance, oxidation of deeper peat due to pneumatophore (breather roots) of tropical trees, and the greater water table fluctuations as a result of the higher hydraulic conductivity of wood peats and tropical climates (Lawson et al.,

2014). In addition, tropical peat is formed as riparian seasonally flooded wetlands with water coming from upstream river networks, whereas the TOPMODEL equations used here implicitly assume a peatland is formed in a grid cell only from rainfall water falling into that grid-cell.".

The methane module was not activated in this study because it has not been updated and evaluated since many years. We informed readers that methane emissions are not resolved by the model on Line484-485: "$CH_4$ and dissolved organic carbon (DOC) are not yet included in the model, both of them are significant losses of C from peatland (Roulet et al., 2007).". And then on Line660-661, we recalled the necessity of including methane and DOC emissions from peatland to draw a more complete picture of peatland C budget: "Including $CH_4$ emissions and leaching of DOC will be helpful to get a more complete picture of peatland C budget.". Actually, in parallel with this study, two projects are ongoing in our group to model $CH_4$ and DOC fluxes from northern regions with ORCHIDEE.

The model resolves one energy budget for all soil tiles in one gridcell, with soil thermal properties of the gridcell being defined as a weighted average of mineral and organic soil (organic soil fraction is prescribed from NCSCD in permafrost regions and from HWSD in non-permafrost regions) (Guimberteau et al., 2018, GMD). In the model, dynamics of peat vs. mineral soil will only affect soil temperature (and permafrost) indirectly: changes of peat vs. mineral soil in the grid cell impacts gridcell soil water content, then gridcell soil water content and water filled fraction of pores impact fusion and solidification heat fluxes in the soil; changes in soil moisture and its liquid/ice state also impact soil thermal conductivity.

Calibrated by the CW-WTD wetland map (Sect. 2.2.1), we compared simulated maximum inundation area of the Northern Hemisphere with CW-WTD in Sect. 4.2 (Fig. S7), on Line404-411. Now, Following the reviewer's suggestion, we move Fig. S7 to the main text. CW-WTD can't be used to evaluate timing of inundation because CW-WTD is a static wetland map. Therefore, in the following figure, we use GIEMS to evaluate inundation timing (Prigent et al., 2007 and 2012, JGR). Note that because wetland extent in GIEMS (the maximum wetland area for the northern hemisphere over 1993-2007 being ~7 million $km^2$ , with lakes are included) are much smaller than in CW-WTD (~13.2 million $km^2$ after excluding lakes) (Tootchi et al., 2019, ESSD), we normalize the data by dividing the simulated and observed total inundated area of each month by the simulated and observed maximum monthly value, respectively, to highlight seasonality of inundation rather than comparing absolute values. Accordingly, the following discussion is added on Line411: "…The model generally captures the spatial pattern of wetland areas represented by CW-WTD (Fig. 5). The multi-sensor satellite-based GIEMS dataset (Prigent et al., 2007, 2012) which provides observed monthly inundation extent over the period of 1993 − 2007 is used to evaluate simulated seasonality of inundation. Fig. 6 shows that the seasonality of inundation is generally well captured by the model, although simulated seasonal maximum of inundation extent occurs earlier than observations (except in WSL) and simulated duration of inundation is longer than observations.".

[Figure]

Fig. 6. Simulated and observed (GIEMS, (Prigent et al., 2007, 2012)) mean seasonality (averaged over 1993–2007) of total inundated area. Note that the simulated and observed total inundated area of each month is divided by the simulated and observed maximum monthly value, respectively, to highlight seasonality of inundation rather than comparing absolute values of inundated area.

**Q3. * The simulated distribution of the peatland area fraction (Fig. 4) shows that the model is able to broadly capture the observed pattern, except that is quite strongly underestimates the peatland extent in the Hudson Bay Lowland (HBL). This reminds me of my own work, where the first version of my model (DYPTOP, ST14) also failed to simulate very high peatland area fractions (over 90%) across this large region. The HBL is, next to the West Siberian Lowland, the largest peatland region and therefore warrants special attention. The failure of the model by Qiu et al. in simulating large peatland fractions may be related to what one may call the "sponge-feedback" – the high efficiency of organic soils in retaining water (small runoff) which in turn increases persistency of flooding and the suitability for peat to accumulate - a positive feedback. I solved this by having (gridcell average) soil parameters that determine the soil hydrology depending on the internally simulated peatland area fraction, rather than using externally prescribed parameters from soil maps. I see that in the present model, some soil parameters are indeed prescribed for each gridcell separately from external data (soil bulk density, soil C fraction; l. 499). I would say that they should be affected by whether the model simulates peatland in the respective gridcell or not. This might be something worth looking into in order to better reproduce the observed Hudson Bay Lowland peatland area fractions. On l.131, it's mentioned that soil thermal (and hydrological?) properties are a weighted average of**

**mineral and organic soils, where organic soil fraction is prescribed from an external dataset (NCSCD and HWSD).**

Actually, the "sponge-feedback" was considered in the present model. In the model, each grid cell is divided into four independent sub-grid hydrological soil unit (HSU): one for bare soil, one for all tree PFTs, one for all short vegetations and one for peatland. The peatland HSU is parameterized with peat-specific hydrological parameters (large porosity, large saturated hydraulic conductivity), while hydrological parameters of other non-peatland HSUs are determined by the dominant soil texture (Coarse/Medium/Fine) of the grid cell. This is described on L114-120: "ORCHIDEE-PEAT version 1 was evaluated and calibrated against eddy-covariance measurements of $CO_2$ and energy fluxes, water table depth, as well as soil temperature from 30 northern peatland sites (Qiu et al., 2018). Parameterizations of peatland vegetation and water dynamics are unchanged from ORCHIDEE-PEAT version 1: …… Vertical water fluxes in peatland tile is modelled with peat-specific hydraulics (Text S1 in the Supplement)."

As for the underestimation of peatland extent in the Hudson Bay Lowland (HBL), Glaser et al. (2004a and 2004b, Journal of Ecology) and Packalen et al. (2014, nature communication) proved that climate alone couldn't explain the initiation and development of peatlands in the HBL, the glacial isostatic adjustment is a more fundamental control of HBL peatlands development. We add sentences on Line434 to address this issue: "……, though the  world's second largest peatland complex at the Hudson Bay lowlands (HBL) is underestimated and a small part of the northwest Canada peatlands is missing. Packalen et al. (2014) stressed that initiation and development of HBL peatlands are driven by both climate and glacial isostatic adjustment (GIA), with initiation and expansion of HBL peatlands tightly coupled with land emergence from the Tyrrell Sea, following the deglaciation of the Laurentide ice sheet and under suitable climatic conditions. The pattern of peatlands at southern HBL was believed to be driven by the differential rates of GIA rather than climate (Glaser et al., 2004a, 2004b). More specifically, Glaser et al. (2004a, 2004b) suggested that the faster isostatic uplift rates on the lower reaches of the drainage basin reduce regional slope, impede drainage and shift river channels. Our model, however, can't simulate the tectonic and hydrogeologic controls on peatland development. In addition, the development of permafrost at depth as peat grows in thickness over time acts to expand peat volume and uplift peat when liquid water filled pores at the bottom of the peat become ice filled pores (Seppälä, 2006). This process is not accounted for in the model and may explain why the HBL does not show up as a large flooded area today whereas peat developed in this region during the early development stages of the HBL complex.".

Unlike the configuration of the model for hydrology, which calculates water budget for each HSU independently. The model can only calculate one energy budget for all HSUs in one grid cell, soil thermal properties are indeed a weighted average of mineral and organic soils (with organic soil fraction being prescribed from NCSCD and HWSD) (Guimberteau et al., 2018, GMD).

**Q4. \* The explicit depth-dependence of the turnover rates is a bit obscure to me. While the rationale is defensible (l. 160 "priming effects, sorption of organic molecules to mineral surfaces"), it's not clear how important this factor is for the simulations here. Couldn't it be avoided? What's the e-folding scale in Eq. 2? (I see that the z_0 parameter is given later in the manuscript) And shouldn't this be accounted for by oxygen conditions, being subject to water content in different layers where the bottom layers will tend to be water-logged and thus have a very low turnover rate. From text S3, this is not evident.**

We understand that in reality, bottom layer of peatland tends to be water-logged and water content of upper soil layers change with time due to the fluctuating water table. While the model resolves water diffusion between soil layers according to the Fokker–Planck equation, shapes of simulated soil moisture profiles depend on soil texture (hydrological parameters), amount and frequency of water input (snowfall, rainfall, runoff from non-peat soils) and water output (evaporation, transpiration, sublimation), water diffusion rates, etc. The figure below shows daily water inputs to a Sweden peatland (68.0°N, 19.0°E) in year1884 and the simulated daily volumetric water content profile for the peat HSU. Simulated soil water content at bottom soil layers are smaller than that at upper layers from Julian days90 to Julian days140, and bottom layers never reach saturation. So, the water content alone can't represent anoxic conditions of peat soil profile.

[Figure]

**Fig. S13.** (Top figure) Daily water inputs to a Sweden peatland (68.0°N, 19.0°E) in year 1884; (bottom figure) simulated daily volumetric water content profile for the peat HSU.

Without the depth modifier, as shown in the figure below, simulated northern peatlands area will not change (3.9 million km$^2$), but northern peatlands C

stock will be underestimated (only 300PgC). We acknowledge that such kind of approach is somehow too empirical but at this stage we can't avoid it. These limitations were presented on Line602: "……The parameter $z_0$, by contrast, exerts a relatively strong control over C profiles. It is noteworthy that while our model resolves water diffusion between soil layers according to the Fokker–Planck equation (Qiu et al., 2018), simulated soil moisture does not necessarily increase with depth (Fig. S13). $z_0$ is therefore an important parameter to constrain peat decomposition rates at depth. With smaller $z_0$, decomposition of C decreases rapidly with depth, resulting in deeper C profile (Fig. S14). Regional scale tests verified these behaviors of the model: When $f_{th}=0.9$ is used (instead of $f_{th}=0.7$), changes in peatland area and peat C stock are negligible (Fig. S15). Without $z_0$, simulated northern peatlands area will not change (3.9 million km2), but northern peatlands C stock will be underestimated (only 300PgC). If $z_0=0.5$ m is applied (instead of $z_0=1.5$ m), the simulated total peat C would triple while the total peatland area would only increase by 0.2 million km2 (Fig. S16)."

[Figure]

(Top figure) Simulated peatland area fraction without the depth modifier ($z_0$), and (bottom figure) simulated peatland soil carbon density without $z_0$.

**Q5. * Comparison with cores. I am not sure if the model presented here can be compared to peat cores. The reason is that, in order to conserve C mass, an expansion of the peatland area fraction has to imply a reduction of the peat C mass per unit area - peat C is effectively diluted over an increasing area. Hence, the vertical growth of peat should slow upon lateral expansion. This is implied by the simplification that the model doesn't explicitly simulate the horizontal dimension. In reality, a peatland has substantial lateral structure and tends to be deep and have the oldest layers towards the center. That's also where peat cores are commonly taken (in order to maximise the temporal coverage). I am**

**therefore not surprised to see that the model appears to generally underestimate peat depth. I suspect that separate simulations are required for this, where the peatland area fraction is held constant (no dilution!).**

We agree with the reviewer that the expansion of peatland area fraction may dilute simulated peat C. Following the reviewer's suggestion, we run simulations with fixed peatland area fraction (with peatland area fraction in each grid cell being derived from the map of Yu et al. (2010, GRL)). However, as shown in the figure below, simulated peat C profiles with varying peatland area fraction (S0 in red line) match better (than S1 with fixed peatland area fraction, in blue line) with observations (black line). This can be due to: (1) in S0, the simulated peatland area fraction is quite small at first, and then it increases gradually. As we add surface runoff of all non-peatland soils in the gridcell into peatland, with a smaller peatland area fraction, S0 tends to create wetter peat soils than S1. (2) in S0, peatland encroach C from non-peatland soils when expanding, and the C is protected from oxic decomposition subsequently. Point (2) will be presented in a follow up study (Qiu et al., in prep). The below figure is not added in the revised manuscript, for simplicity, we add a section in discussion on Line543 in the revised manuscript to note the dilution issue:

**Vertical profiles of peatland soil organic carbon**

We note that caution is needed in interpreting the comparison between simulated peat C profile and measured C profile from peat cores (Fig. 3, Fig. 4). In reality, peat grow both vertically and laterally since inception, with the peat deposit tend to be deeper and its basal age tend to be older at the original nucleation sites / center of the peatland complex (Bauer et al., 2003; Mathijssen et al., 2017). As mentioned earlier, field measurements tend to take samples from the deeper part of a peatland complex and shallow peat are underrepresented. The model, however, only simulates peat growth in the vertical dimension and lacks an explicit representation of the lateral development of a peatland in grid-based simulations, thus simulated peat C (per unit peatland area) is diluted when the simulated peatland area fraction in the grid cell increases.

[Figure]

**Q6. \* The authors aim to model peat C dynamics during the Holocene (see title), but relatively little focus is given to forcing and evaluating the model with respect to this palaeo perspective. As far I understood, the model is forced with constant pre-industrial climate (although insolation and summer temperatures varied substantially during the Holocene, especially at high latitudes). Was a changing sea level accounted for? For applications in palaeo climate and -carbon cycle studies, the model is expected to reliably simulate the net C balance of peatlands. I am not convinced that the evaluation of C content across the soil profile, as presented in the paper, provides sufficient information to evaluate this aspect. Shouldn't a comparison be done against dated peat cores, where the amount of C (left today) per age bin is given? The model doesn't track age bins explicitly, but could be extended to simulate C14 decay and transport across the soil layers (so that lower layers would have an older C14 age, which could then be compared to the C14 age across depth in dated cores). Alternatively, one could write out soil C inputs and decomposition rates at all time steps and resolve age cohorts explicitly offline (diagnostically). I understand that this is a substantial challenge, but I am not fully convinced that the evaluation presented here is sufficient. At least a discussion of these points should be added.**

Considering that there are significant variations in both proxy-based reconstructions of Holocene climate and climate models simulated Holocene climate by models, and a significant model-data discrepancy exists (Mann et al., 2008, PNAS; Liu et al., 2014, PNAS), we simply used looped 1961-1990 climate in this study to approximate the higher Holocene temperatures relative to the 'pre-industrial' period (Marcott et al., 2013, Science). Uncertainties induced by the climate forcing has been discussed on Line532-540, and one of our future work is to study impacts of different Holocene climate forcing data.

ORCHIDEE is a land surface model simulating $CO_2$, water and energy fluxes of terrestrial ecosystems. It is the land component of the IPSL-CM5 (Atmosphere-Land-Ocean-Sea ice) earth system model. In this study, ORCHIDEE-PEAT was run offline, sea-level changes were not accounted for. But changes in the exposed land area after the retreat of ice sheet were considered (see Sect. 3.2).

The reviewer is right that we can't compare simulated peat C profile against dated peat cores because our model doesn't track age bins explicitly. Tifafi et al. (2018, GMD) incorporated 14C dynamics in the soil into the ORCHIDEE-SOM model. Their work is in parallel with our model, but could be merged together in the future developments. A discussion on this issue is added following Q5: "......, thus simulated peat C (per unit peatland area) is diluted when the simulated peatland area fraction in the grid cell increases. In addition, while a dated peat core tells us net burial of peat C during time intervals, the model can't provide a peat age-depth profile because it simulates peat C accumulation based on decomposition of soil C pools, rather than tracking peat C as cohorts over depth/time (Heinemeyer et al., 2010).

The above-noted discrepancies between the simulation and the observation highlight both the need for more peat core data collected with more rigorous sampling methodologies and the need to improve the model. In parallel with this study, [14]C dynamics in the soil has been incorporated into the ORCHIDEE-SOM model (Tifafi et al., 2018), which may give us an opportunity to compare simulated [14]C age-depth profiles with dated peat C profiles in the future after being merged with our model.".

**Q7. * I simply did not understand Fig. 1.**

Fig.1 was used to show the RMSE of simulated and measured peat depth at 60 peatland sites. The transition from green to white indicates the measured mean depth of all 60 sites. Since information showed by this figure was already informed by Table1, we replaced it with a figure for modelled vs. observed depth at these sites. And the text from Line369 to Line 372 were modified accordingly: "...... Peat depths are underestimated for most sites (Fig. 1). Simulated depth of these 57 sites ranges from 0.37 m to 6.64 m and shows a median depth of 2.18 m, while measured peat depth ranges from 0.96 to 10.95 m, with the measured median depth being 3.10 m (Table 1). The root mean square error (RMSE) between observations and simulations is 2.45 m."

[Figure]

Fig. 1. Measured and simulated peat depth at 60 peatlands sites (Table S1). Shapes of markers indicate peatland types (bogs, fens, others), colors of markers imply climatic zones (temperate, boreal, arctic) of sites' location.

**Q8. * Should become clear upfront what parameters are calibrated and what observational targets are used for calibration.**

We add the following table to show parameters used and calibrated in the model. We generally described the model parameter, its calibration and observational target simultaneously in the main text. To retain readability (and not having to add redundant descriptions), we keep these descriptions as presented in the initial manuscript.

Table 1. Parameter values in peatland modules of ORCHIDEE-PEAT v2.0.

| Parameter | Value | Description |
|---|---|---|
| $k_{0,i}$ | | the base decomposition rate of carbon pools, Eq. 1 |
| $k_{0,i}$ : $i=active$ | 1.0 $a^{-1}$ | the base decomposition rate of the active pool at 30 °C, Eq. 1 |
| $k_{0,i}$ : $i=slow$ | 0.027 $a^{-1}$ | the base decomposition rate of the slow pool at 30 °C, Eq. 1 |
| $k_{0,i}$ : $i=passive$ | 0.0006 $a^{-1}$ | the base decomposition rate of the passive pool at 30 °C, Eq. 1 |
| $z_0$ | 1.5m | The e-folding depth of base decomposition rate, Eq. 2 |
| $f$ | 0.1 | The fraction of carbon content in the model layer to be transported to the layer below, Eq. 4 |
| $f_{th}$ | 0.7 | The amount (fractional) of carbon content that the model layer need to hold before transporting carbon to the layer below, Eq. 5 |
| $m$ | gridcell specific | TOPMODEL parameter (the saturated hydraulic conductivity decay factor with depth), Fig.1, TextS4 |
| $CTI_{min}$ | gridcell specific | TOPMODEL parameter (the minimum CTI for floodability), Fig.1, TextS4 |
| $Num$ | gridcell specific | The total number of growing season months in the preceding 30 years, Fig.1, Sect. 2.2.2 |
| $SWB$ | 6 cm | Minimum summer water balance, Fig.1, Sect. 2.2.2 |
| $C_{lim}$ | 50.3 kg C $m^{-2}$ | Minimum peat C density , Fig.1, Sect. 2.2.2 |

**LESS MAJOR (BUT NOT MINOR)**

**Q9. * Better define the scope of the model and the evaluation, the scale at which the model is expected to yield reliable results, what simplifications have taken to get there, and where the model is not applicable. This can be achieved by more clearly stating upfront for what research questions the model is expected to be applied, and what it therefore needs to simulate with fidelity (and why these quantities). And then present the results with a focus and structure to address these quantities. This is largely done so already, but it would greatly help the reader to improve the structure of the paper in this sense. I would expect the following key quantities:**
**\* total (northern) peat C: ok**
**\*spatial patterns of peatland extent: ok, although the particularly extensive peat area in the Hudson Bay Lowland is largely missed by the model.**
**\* basal age/inception, compared to first year of peatland establishment in model: It would be good to evaluate simulated and observed basal ages across space, e.g. with a map showing the simulated basal age across space and dots on top of it for observed basal ages from different cores.**
**\* peat C accumulation/respiration history: The net C balance through time is what is relevant for the C cycle (what the atmosphere "sees"). I am not convinced that the evaluation presented here, looking at C content across depth, is giving us the right information to evaluate the model in this respect. The dimension time is missing (as mentioned above); there is no age scale of the cores factored into the analysis.**

We revise the summary on Line657-Line661 to make the scope of the study clearer for readers (not stating upfront because we don't want to cause confusion to readers, they need to get an idea of the model and the simulation protocol first): "As a large-scale LSM which is designed for large-scale gridded applications, ORCHIDEE-PEAT v2.0 cannot explicitly model the lateral development of a peatland. The model therefore aims to simulate average peat depth and C profile in a grid location rather than capturing peat inception

time and age-depth profiles of peat cores. For tropical peatlands, the model needs to be improved to represent its tree dominance, oxidation of deeper peat due to pneumatophore (breather roots) of tropical trees, and the greater water table fluctuations as a result of the higher hydraulic conductivity of wood peats and tropical climates (Lawson et al., 2014). In addition, tropical peat is formed as riparian seasonally flooded wetlands with water coming from upstream river networks, whereas the TOPMODEL equations used here implicitly assume a peatland is formed in a grid cell only from rainfall water falling into that grid-cell. Further work to improve this simulation framework is needed in areas such as an accurate representation of the Holocene climate, higher spatial resolution, distinguish bogs from fens to better parameterize water inflows into peatland. Including $CH_4$ emissions and leaching of DOC will be helpful to get a more complete picture of peatland C budget.".

Questions concerning these quantities had already been disclosed previously (or later) by the reviewer, please see our responses to Q3, Q6 and Q13:

*spatial patterns of peatland extent in the HBL – Q3
* basal age/inception – Q13
* peat C accumulation/respiration history – Q6

**Q10. * Vertical peat growth model: I didn't intuitively understand the rationale for using bulk density data to formulate the vertical growth/downward transport model. Why didn't you use volumetric C content? Can your approach be described as a sequence of C-buckets that fill up by receiving inputs from the layer above (once this "spills over")? Then, spill-over is happening when the typical empirical volumetric C density at the respective depth, as measured in your 102 cores, is achieved. I'm just thinking out loud here, trying to make sense of the model. But maybe you can include such an intuitive description of your approach in the paper.**

Actually, we did use volumetric C content in the vertical downward transport model. In Eq.3 (on Line182), we used observed bulk density and observed C concentration (%) to calculate an empirical amount of C (kg C/$m^2$) that each model layer can hold ($M_l$), then simulated C content is compared with $f_{th}*M_l$ ($f_{th}$ is a prescribed value, $f_{th}$ = 0.7) to start downward transport of C. The reviewer's description generally matches our initial idea, although we calibrated the threshold to start downward C transfer and the amount of C to be transferred according to peat cores.  Please see our responses to Eq. 4 and Eq. 5.

**Q11. * While the striking performance in simulating inundation is definitely a plus, it remains unclear how this improvement over earlier publications (e.g., ST14) is achieved. Is it related to resolving the soil hydrology across layers instead of using a simple bucket model? The inundation sub-model is key for the peatland extent model and warrants a bit more attention in the paper.**

We would like first to note that the soil hydrology scheme is not the only difference between our model and ST14. Representation of peatland vegetations, the soil thermal regime, and snow processes are all very different

between the two models (Guimberteau et al., 2018; Stocker et al., 2014, GMD; Sitch et al., 2013, GCB; Ekici et al., 2015, The Cryosphere). All the above-mentioned processes can more or less have an influence on both the water fluxes and water content of a grid cell, and also affect simulated inundation.

In addition, we use peat-specific soil hydrological parameters for peatland, while using another set of parameters (which depend on the dominate mineral texture of the grid cell) for mineral soils. In contrast, ST14 used grid cell average soil parameters in soil hydrology. De Rosnay et al. (2002, JGR) evidenced that a single "average" textured soil couldn't adequately represent the "averaged" water fluxes for heterogeneous regions, a subgrid-scale representation of soil type is relevant for modeling of soil water movement and surface fluxes.

For reasons indicated above, we feel that a comparison between ST14 and our model would be unfair, and we couldn't attribute the better performance of our model in simulating inundation than ST14 only to the multi-layer physically-based soil hydrology.

As for impacts of the 2-layer bucket scheme vs. the 11-layer physically-based diffusion scheme, while a comparison between the two schemes is out of the scope of this paper, we can get an idea of it from the study of De Rosnay et al. (2002, JGR) and Guimberteau et al. (2014, GMD). De Rosnay et al. (2002, JGR) showed that compared to the 2-layer bucket approach, the multi-layer diffusion scheme together with a subgrid-scale representation of soil type allow a more realistic representation of surface water fluxes, soil moisture profile and root water uptake, resulting in a better spatial and seasonal representation of evapotranspiration. Guimberteau et al. (2014, GMD) applied both the simple 2-layer scheme and the 11-layer diffusion scheme over the Amazon Basin, and showed that the 11-layer diffusion scheme simulates more dynamic soil water storage variation and improves simulation of soil water storage when compared with satellite observations.

**Q12. * I don't think it's appropriate to require every model presented in GMD to be fundamentally novel. Furthermore, the model presented by Qiu et al. is largely an adoption of an existing model (ST14), which itself is based on Kleinen et al., 2012. Sufficient reference is made by Qiu et al. to this earlier work. However, the authors introduce and motivate their work with (l.94) "While both studies made pioneering progresses in the modelling of peatland ecosystems, they adopted a simple bucket approach to model peatland hydrology and peatland C accumulation, and neither of them resolved the diel cycle of surface energy budget." However, it is unclear why the diurnal surface energy budget needs to be explicitly simulated in this context, and what limitations the simple bucket model approach incurs. It definitely needs more clarification what the model adds to our knowledge and our predictive power and I am skeptical that resolving the diurnal cycle of surface energy exchange adds a great deal. I am more curious about whether resolving soil hydrology across multiple layers helps better simulating relevant peatland-related processes, but the paper doesn't provide this insight. I think it is**

**important that it becomes better clear what the merit of this model (over existing ones) is.**

Actually, the diurnal cycle of surface energy exchange matters. We ran two simulations (with diel cycle vs. without diel cycle) to show the model performance in simulating peatland NEE and C stock at the Degerö Stormyr peatland site (Peichl et al., 2014). ORCHIDEE-PEAT v2.0 resolved energy processes in 30min time steps. In the first simulation (with diel cycle), we used measured half-hourly meteorological variables from the flux tower to force the model; in the second simulation, to mimic a run without diel cycle of meteorological variables, the daily mean of measured meteorological variables is used. In the figure below, observed peatland NEE (negative NEE: $CO_2$ sink) of the site in 2002 is shown in black, simulated NEE with diel cycle is shown in red, and simulated NEE without diel cycle is shown in yellow. Simulated NEE with diel cycle matches better with observations. Meanwhile, in the simulation without diel cycle, simulated C density is 50% greater than the simulation with diel cycle.

Regarding impacts of the multi-layer, physically-based soil hydrology scheme, please see our responses to Q11

[Figure]

**Q13. * Observed (Mc Donald et al., 2006) and modelled inception age could be compared across space rather than just showing the numbers across time in Fig. 10. Actually, this comparison is subject to a possible sampling bias in Mc Donald. You want to test whether the model simulates the right inception time at a specific location, and not only the fraction of total number of simulated against the total number of sampled peatlands sampled in each age bin.**

We couldn't transiently run the model due to the limitation of computational resources, so we spun up the model at discrete Holocene intervals with the soil C only part of the ORCHIDEE LSM being forced by archived litter input from a 100 years simulation with full ORCHIDEE (2000 yr each time) in this

study. In other words, we first calculated peatland areas at 12,000 BP (Area0), then we assumed that peatland areas will not change in the following 2000 years, and we simulated C accumulated by Area0 in the following 2000 years. Then we updated peatland areas (Area1, at 10,000 BP) and simulated C accumulated by Area1 for another 2000 years…… As a result of this crude spinup acceleration procedure, we aimed to reproduce peatland areas and peat C stocks at discrete Holocene intervals rather than to capture inception time of peat cores. On the other hand, the inception time of peat cores couldn't represent the area development of a peatland, i.e. the simulated first initiation of peatland in a specific grid cell could be quite early, while the lateral expansion occurred much later. As the model was simulating peat C accumulation based on peatland areas (Area0, Area1…) at discrete Holocene intervals, we feel that the comparison at 2000-yr age bins is informative.

**MINOR**

**\* l.21: I wouldn't subscribe to 'recently'.**

Deleted now from the text.

**\* l.34: "270-540 PgC" Seems to be at the low end. What's the reference? On l. 44 references are given. But I suggest to use the latest (Yu, 2010) as the benchmark.**

We thank the reviewer for the suggestion. Considering that there are still large uncertainties in peat C stock estimates (Yu et al., 2012, Biogeosciences) and there is still no consensus in the soil (peat) science community, we feel that using only one benchmark is not rigorous enough (although the estimate by Yu (2010) is indeed the latest). We decide to report the range of peatland C stock estimates, as presented in the initial manuscript.

**\* l.48 "in environments…" Make a new sentence, as this is not related to the first part of the sentence**

We rephrase this sentence as: "Due to water-logged, acidic and low-temperature conditions, plant litter production exceeds decomposition in northern peatlands. More than half of northern peat carbon was accumulated before 7000 years ago during the Holocene (Yu, 2012)."

**\* l.49: Change 'despite' to 'while'.**

Changed in the text

**\* l.64/65: Weird sentence. The depth itself doesn't prevent oxygen supply.**

We rephrase this sentence as: "Water table is one of the most important factors controlling the accumulation of peat, because it limits oxygen supply to the saturated zone and reduces decomposition rates of buried organic matter (Kleinen et al., 2012; Spahni et al., 2013)."

**\* l. 69: Unclear: "critical level [of WTD???]"**

We rephrase this sentence as: "…However, some studies showed that changes in soil water content could be very small while the water table was lowering, the drawdown of the water table caused only small changes in soil

air-filled porosity and hence exerted no significant effect on ER (Lafleur et al., 2005; Parmentier et al., 2009; Sulman et al., 2009)."

**\* l. 70: Isn't WTD linearly related to soil moisture content? Why the threshold?**

It is intuitive that WTD is closely related to soil moisture content. However, we were not talking about the total soil moisture content of the peat profile here. We were considering the relationship between WTD and the moisture content of soils above the WT, because oxic respiration above the WT contributes more to the heterotrophic respiration of peat than anoxic respiration below the WT.

The figure below (Figure 2 of the study of Lafleur et al. (2005, Ecosystems)) shows measured WTD and soil water content at the Mer Bleue peatland site (soil water content was measured with a profile of TDR probes). As shown by Figure2a, the soil water content at 0.28m depth decreased rapidly when WTD drops from -25cm to -33cm, however, it only decreased marginally with further drops of WTD (from -33cm to -70 cm).

[Figure]

Figure 2. Relationship between peat moisture content as a percent by volume and water-table depth, WTD, below the hummock surface at **A**) −28 cm soil depth and **B**) −48 cm soil depth.

**\* l.69-74: This sounds like the authors highlight a unresolved challenged here that the model/paper is going to address. However, it's unclear what is meant here (of course, WTD determines soil moisture or vice-versa),**

**and how the model and results presented here address this particular challenge.**

Yes, here we highlighted the fact that ecosystem respiration didn't always depend on WTD, there could be only small changes in soil moisture in the unsaturated part of the peat profile while WTD was significantly lowered (Lafleur et al., 2005, Ecosystems; Parmentier et al., 2009, Agric. For. Meteorol.; Sulman et al., 2009, Biogeosciences). Founded on a physically-based representation of hydrology of our model, the decomposition of peat C at each model layer is controlled by peat soil moisture (the soil volumetric water content - respiration relationship for organic soils from the meta-analysis of Moyano et al. (2012, Biogeosciences) were used), rather than by WTD.

We haven't ran a control simulation with decomposition controlled by WTD, and the aim of this study was to evaluate if the model can reproduce present-day peatland areas and C stocks, thus we didn't address this in the results. This particular issue (two-layered model vs. multi-layered model, WTD controlled vs. moisture controlled decomposition) can be addressed in future studies.

**\* l.76: Style: don't refer to 'groups'.**

"groups" is deleted in the text.

**\* l. 92: I would say that the key in ST14 was to account for peatland-specific water storage capacity in typical organic soils ("sponge" feedback) which enabled to accurately simulate the particular patter of peatland areas across the globe.**

We add a sentence to highlight this key improvement on Line92: "Stocker et al. (2014) extended the scope of Kleinen et al. (2012) in the LPX model. In their model, soil water storage and retention were enhanced and runoff was reduced by accounting for peatland-specific hydraulic properties. A positive feedback on the local water balance and on peatland expansion was therefore exerted by peatland water table and peatland area fraction within a grid cell. Areas that are suitable for peatland development were distinguished from wetland extent according to temporal persistency of inundation, water balance and peatland C balance."

**\* l. 98: Unclear what "discrepancies" are referred to.**

Here, "discrepancies" referred to issues mentioned above; On Line54-Line74: decomposition doesn't always depend on WTD, soil moisture controls decomposition. On Line75-85: vertical heterogeneities in soil temperature, moisture and soil freezing can't be captured by two-layered bucket model. On Line86-Line97: previous models used two-layered bucket approach to model peatland hydrology and C decomposition without diel cycle of energy and water budget. To keep it as simple as possible (and not having to add redundant descriptions), we only rephrase this sentence on Line98 as: "To tackle these above-mentioned discrepancies and estimate the C dynamic as well as the peat area, we used the ORCHIDEE-MICT land surface model incorporating peatland as ……".

**\* l.121: 'multi' instead of 'many'**

Corrected in the text.

**\* Eq. 4: Why isn't it flux = f \* (C_l - M_th,l)? The way it's formulated, the C content may drop below the threshold after transfer. Shouldn't it stay "saturated" after accounting for downward transport?**

This question and the question following this one are actually asking the same question: why didn't we keep the soil layer "saturated" after downward C transfer. Please see our responses to the question following this one.

**\* Eq. 5: What's the rationale for introducing parameter f_th? Why isn't it 1?**

We calibrated $f$ and $f_{th}$ in Eq.4 and Eq.5 to match the simulated vertical C profiles with peat cores. Actually, we have also tried to formulate the flux as flux = $C_l$-$M_l$ so that the layer stay "saturated" after C transfer, in this case, simulated vertical C profiles in site level simulations don't match with peat cores as well as with Eq. 4 and Eq5; and in regional simulations, the simulated peatland C in West Siberia and southeastern US are worse than with Eq.4 and Eq.5 (see the figure below). The formulations of the downward C transfer model will be tested for the next steps of model application and development.

[Figure]

**\* I. 216: …than what? Explanation would be helpful: More computationally efficient than determining water table depth for each sub-grid pixel.**

We improve the sentence on Line216 as suggested: "……, which is more computationally efficient than determining water table depth for each sub-grid pixel (Stocker et al., 2014)."

**\* I. 227: Ambiguous formulation. Do you mean max(monthly values)?**

Yes, we meant max(monthly values). We correct the text on Line 227 as: "…… We therefore compare simulated maximum monthly mean wetland extent over 1980−2015……."

**\* Section 2.2.2.: Put Fig. S2 into main text and highlight difference to ST14.**

We move Fig. S2 into the main text in the revised manuscript, its difference to ST14 is highlighted on Line284: "The difference between our model and the DYPTOP model in simulating peatland area dynamics can be summarized as follows: (1) TOPMODEL calibration: TOPMODEL parameters are globally uniform in the DYPTOP model, but grid cell-specific in ORCHIDEE-PEAT v2.0. (2) Criteria for peatland expansion: In the DYPTOP, the "flooding persistency" parameter is globally uniform, being 18 months in the preceding 31 years. And the ecosystem water balance is expressed as annual precipitation-over-actual-evapotranspiration (POAET). In ORCHIDEE-PEAT v2.0, the flooding persistency parameter is grid cell-specific, being the total number of growing season months in the preceding 30 years. And peatland expansion is limited only by summer water balance. The relative areal change of peatland is limited to 1% per year in DYPTOP, but not limited in our model. (3) Peatland initiation: DYPTOP prescribes a very small peatland area fraction (0.001%) in each grid cell to simulate peatland C balance condition. Peatland can expand from this "seed" once water and carbon balance criteria are met. In ORCHIDEE-PEAT v2.0, no "seed" is needed because only the flooding persistency and summer water balance criteria need to be met for the first initiation of peatland (Fig. 1b), carbon balance is only checked after initiation (Fig.1c)."

**\* l. 238: Not quite correct. I don't know what the authors refer to here.**

We revised the sentence on Line238 as: "Stocker et al. (2014) introduced a 'flooding persistency' parameter (N in Eq.12, Eq.13 in Stocker et al. (2014)) for the DYPTOP model to represents the temporal frequency of inundation. N is a globally uniform parameter in DYPTOP, being set to 18 months during the preceding 31 years."

**\* l. 249: Question: Are non-growing season months discarded or do they count towards N?**

Non-growing season months are counted.

**\* l. 254: What's the difference to ST14? Using only water balance during summer months instead of entire year? Might be worth mentioning explicitly. What's the rationale for this choice? Note that winter precipitation is relevant too as summer snow melt is effectively delayed winter precipitation.**

Yes, only water balance during summer months are used in the ORCHIDEE-PEAT v2.0 model. This difference to ST14 now is added on Line284 in the revised manuscript (please see our response to the previous question regarding Section 2.2.2.). Summer dryness was proved to be a key factor in limiting *Sphagnum* growth and peatland expansion in western Canada (Gignac et al., 2000, Journal of Biogeography) and in Western Siberia (Alexandrov et al., 2016, Scientific Reports). Based on the abundance of *Sphagnum* species on 640 peatland sites located in western Canada, Gignac et al. (2000) evidenced that *Sphagnum*-dominated peatlands do not occur in areas having summer moisture index (P–PET) values ≤ -6 cm. A similar climate characteristic, warm precipitation excess (P-0.7PET), was reported by Alexandrov et al. (2016), to explain the present-day distribution of peatlands

in Western Siberia, their absence during the Last Glacial Maximum, and their expansion during the mid-Holocene.

**\* l. 258 "May-September": Warning: this would mean that the model is not applicable in the south.**

A warning is added on Line261: "……SWB = 6 cm is selected so that the model captures the southern frontier of peatland in Eurasia and western North America (Text S5). Note that the definition of summer (May-September) and *SWB* are not applicable for tropical regions and the Southern Hemisphere."

**\* l. 270: Clarify: C_lim "is defined here as…"**

$C_{lim}$ is clarified on Line270 in the revised manuscript: "…… $C_{lim}$ is defined here as long-term peatland C balance condition, it's a product of……"

**\* l. 278: "SubC": What's this?**

SubC is a stand-alone soil carbon sub-model of ORCHIDEE, it simulates only soil carbon dynamics using monthly litter and soil C input, soil water and thermal conditions from the preceding full ORCHIDEE run. This part is described in Sect.3.2, L312-315.

**\* l. 280-283: Add reference to ST14 as this is the same procedure as chosen by them.**

Reference is added in the text: "Therefore, parameterizations of the "old peat" pool is identical to mineral soils, following the study of Stocker et al. (2014). When peatland expansion happens, the peatland will first expand into this 'old peat' area and inherit its stored C (Stocker et al., 2014)."

**\* l. 350: Missing references**

References are added: "Simulated peatlands SOC is evaluated against: 1. The WISE database (Batjes, 2016); 2. The IMCG-GPD (Joosten, 2010)."

**\* l. 371: Figure for modelled vs. observed depth would be instructive.**

A figure for modelled vs. observed depth is added, please see our response to Q7.

**\* Fig 1/Sect. 4.1: I didn't understand Fig. 1 and how I can read the RMSE from that figure. I expected a comparison of modelled and observed peat depth (or total column C), possibly split by temperate/boreal/arctic and/or bog/fen.**

A figure for modelled vs. observed depth is added, please see our response to Q7.

**\* l. 411: Worth including figure in main text.**

The figure (Fig. S7) is moved to the main text now.

**\* l. 425: Are leptosols and agricultural peatlands simply deducted from simulated peatland areas?**

Leptosols and agricultural peatlands were deduced from both simulated areas and simulated C stocks. To clarify it, we modify the sentence on Line425: "After masking Leptosols and agricultural peatlands from the simulated peatland areas and peatland C stocks, ……"

**\* l. 440: Abbreviations introduced?**

These Abbreviations were introduced on Line357: "……, from literature data on peatlands in North America (NA) and in the West Siberian lowlands (WSL)."

**\* l. 455: "we can only make the..." I don't understand this part.**

We couldn't do transient spinups due to the limitation of computational resources with the full ORCHIDEE LSM, so we designed an accelerated multiple spin-up strategy (Sect. 3.2): For regions that were unglaciated during Holocene, we ran SubC (a stand-alone soil carbon sub-model that only simulates soil C dynamics, without having to run the full ORCHIDEE) for 6 times, with SubC simulates C decomposition and accumulation over 2000 years each time. Therefore, we only know simulated peat depth at 2000-year intervals in regional simulations, and we can only make the comparison (observed vs. simulated peat depth) at 2000-year intervals. For example, for a peat core with its age being 8500 years, we compare its observed depth with simulated peat depth after the fourth SubC run.

**\* l. 489: "several": delete**

Deleted.

---

## Author Comment (AC2) · 28 Mar 2019

Response to Referee #2's comments

We would like to thank Benjamin Stocker and the anonymous referee very much for their constructive comments. In the following, please find our point by point response to the comments.

- Reviewer's comments are in bold
- Modifications done in the revised manuscript are in blue
- All figure numbers, table numbers, and line numbers refer to the initial manuscript version.

**Anonymous Referee #2**

**Qiu et al. present their new peatland model, ORCHIDEE-PEAT (v2) and use it prognostically simulate peatland C, extent, and depth over the Holocene. Their work borrows from previous efforts using TOPMODEL based approaches but they extent the field by allowing their model to determine where peatlands will initiate and expand. I find the work to be on the whole sound and interesting. The problem they are tackling is far from trivial and I am surprised it does as well as it does. I am a little concerned about the poorer performance in the major peatland complexes of the world (Hudson's Bay and West Siberia) which I get to in my comments. The paper is generally easy to follow and has relatively few typographical/grammatical errors. I think the paper is publishable in GMD but would like to see my comments addressed prior to that.**

**Main comments:**
**1. The paper seems to sometimes confuse wetlands and peatlands. While peatlands are a type of wetland, in the paper the distinction can be at times very fuzzy. For example, in the abstract it says 'A cost-efficient TOPMODEL approach is implemented to simulate the dynamics of peatland area, calibrated by present-day wetlands areas that are regularly inundated or subject to shallow water tables' (lines 28 - 30). Since it is very possible to have a non-peatland wetland be 'regularly inundated or subject to shallow water tables' this makes it confusing at a minimum. Later in the supplementary material some model parameters are tuned, grid cell by grid cell, to 'select the combination that matches with the CW-WTD wetlands map'. So it appears quite unclear that this is indeed a peatland specific parameterization. I realize that there are other steps to determine if peat will begin to form at the site (e.g. Fig S2) but the implementation of the wetland/peatland determination scheme is confusing. Why tuned to wetland area if that will include many non-peat wetlands? Is the idea that the peatland initiation scheme can handle the rest? Can the authors try and bring a bit more clarity to that aspect of their technique?**

The reviewer is right that not all wetlands are peatland, non-peat wetland can also be regularly inundated or subject to shallow water tables. In our study, the cost-efficient TOPMODEL was calibrated to reproduce wetland distributions (CW-WTD, which includes non-peat wetlands). Then, based on

the study of Kleinen et al. (2012, Biogeosciences) and Stocker et al. (2014, GMD), we assumed that peatland can be distinguished from other wetland, using the peatland initiation condition and development scheme which includes inundation persistency, summer water balance and long-term C balance criteria.

We appreciate the reviewer's suggestion that the distinction between peatland and wetland should be clearer, we thoroughly checked the manuscript and revised the text where the distinction between them was fuzzy:

On Line28-30: . The cost-efficient version of TOPMODEL and the scheme of peatland initiation and development from the DYPTOP model, are implemented and adjusted, to simulate spatial and temporal dynamics of peatland.

On Line92: Stocker et al. (2014) extended the scope of Kleinen et al. (2012) in the DYPTOP model. In their model, soil water storage and retention were enhanced and runoff was reduced by accounting for peatland-specific hydraulic properties. A positive feedback on the local water balance and on peatland expansion was therefore exerted by peatland water table and peatland area fraction within a grid cell. Areas that are suitable for peatland development were distinguished from wetland extent according to temporal persistency of inundation, water balance and peatland C balance.

On Line102-103:  Peatlands extent are modelled following the approach of DYPTOP (Stocker et al., 2014) but with some adaptions and improvements (Sect. 2.2).

On Line126-131: Furthermore, the  approach proposed by Stocker et al. (2014) is incorporated into the model to simulate dynamics of peatland area (Sect. 2.2). This model simulating the dynamics of peatland extent and the vertical buildup of peat is hereinafter referred to as ORCHIDEE-PEAT v2.0.

On Line205-206:  Here, a cost-efficient TOPMODEL from the DYPTOP model (Stocker et al., 2014) is incorporated, and calibrated for each grid cell by present-day wetland area that are regularly inundated or subject to shallow water tables, to simulate wetland extent (Sect. 2.2.1). Then, the criteria for peatland expansion is adapted from DYPTOP to distinguish peatland from wetland (Sect. 2.2.2).

**2. I fully understand the authors' point about difficulty in simulating small permafrost complexes (e.g. discussion of Fig 6) but I am concerned about the poorer performance in the major complexes such as the HBL or WSL. Both of these regions have areas of near 100% peatland cover so the model should have a good chance. Also there is an overabundance of peatlands in some regions that are generally devoid of peatlands (e.g. E. USA). Is this 'smearing' of peatlands perhaps a result of how wetlands area is generally determined, i.e. TOPMODEL-based, or is this a result of the peat initiation limits? I think this deserves more discussion in the paper as it is a striking aspect of the result and one that the community**

**would benefit from any lessons learned regarding how to best get the hotspots without overdoing the rest of the domain.**

Simulated peatland areas at the WSL (~ 0.6 million km$^2$) matched with observation-based estimates (in PEATMAP: ~ 0.6 million km$^2$; in WISE: ~ 0.5 million km$^2$). But the model indeed underestimated peatland areas at the HBL, and the same question has been raised by Referee1 (his Q3). Below are our responses to the question: As for the underestimation of peatland extent in the Hudson Bay Lowland (HBL), Glaser et al. (2004a and 2004b, Journal of Ecology) and Packalen et al. (2014, nature communication) proved that climate alone couldn't explain the initiation and development of peatlands in the HBL, the glacial isostatic adjustment is a more fundamental control of HBL peatlands development. We add sentences on Line434 to address this issue: "……, though the  world's second largest peatland complex at the Hudson Bay lowlands (HBL) is underestimated and a small part of the northwest Canada peatlands is missing. Packalen et al. (2014) stressed that initiation and development of HBL peatlands are driven by both climate and glacial isostatic adjustment (GIA), with initiation and expansion of HBL peatlands tightly coupled with land emergence from the Tyrrell Sea, following the deglaciation of the Laurentide ice sheet and under suitable climatic conditions. The pattern of peatlands at southern HBL was believed to be driven by the differential rates of GIA rather than climate (Glaser et al., 2004a, 2004b). More specifically, Glaser et al. (2004a, 2004b) suggested that the faster isostatic uplift rates on the lower reaches of the drainage basin reduce regional slope, impede drainage and shift river channels. Our model, however, can't simulate the tectonic and hydrogeologic controls on peatland development. In addition, the development of permafrost at depth as peat grows in thickness over time acts to expand peat volume and uplift peat when liquid water filled pores at the bottom of the peat become ice filled pores (Seppälä, 2006). This process is not accounted for in the model and may explain why the HBL does not show up as a large flooded area today whereas peat developed in this region during the early development stages of the HBL complex.".

As for the overestimation of peatlands in east US, it could be related to past land use change in peatlands. According to the U.S. Fish and Wildlife Service's National Wetlands Inventory (Tiner Jr, 1984; Dahl, 2011), there were about 215 million acres of wetlands in the lower 48 states of US at the time of the Nation's settlement, but only 110 million acres remained by 2009 due to agricultural development, urban and other development (~50% of wetlands in the conterminous US has been lost to land use change). From 1780's to mid-1980's, 6 states lost more than 85% of their wetlands, and 16 states lost 50%-85% of their wetlands (Dahl and Allord, 1997). Although wetlands are not necessarily peatlands, the reported losses of wetlands in US indicating that a potentially large area of peatlands in US may have been lost to land use. However, historical losses of peatlands due to land use change and the impact of agricultural drainage of peatlands haven't been taken into account by our model. Simulated natural peatland area by 1860 is 0.4 million km$^2$, if we assume that 50% of simulated natural peatlands have been lost to land use change (the same percentage of historical wetlands losses) and there is no change in peatland area since then, then ~0.2 million km$^2$

remained as natural peatlands, closer to observation-based estimates (0.05-0.1 million km$^2$).

We add sentences on Line626 to address this issue: "From early 1600's to 2009, ~ 50% of the original wetlands in the lower 48 states of US have been lost to agricultural, urban development and other development (Dahl, 2011; Tiner Jr, 1984). Although wetlands are not necessarily peatlands, the reported losses of wetlands in US indicating that a potentially large area of peatlands in US may have been lost to land use change. However, historical losses of peatlands due to land use change and the impact of agricultural drainage of peatlands haven't been taken into account by our model."

**Minor comments:**
**1. line 202 - does that mean the peatland PFTs are forced into their gridcells? Can you expand on what peatland PFTs there are? I see that there are some mention in Text S1 but it just says a PFT with shallow roots. Is it a tree? Do you simulate any other peatland specific PFTs? Shrubs? Moss? Sedges?**

There is only one peat-specific PFT in this study, it is forced into the gridcell as long as the peatland development criteria are met. This peatland PFT represents an average of all vegetations growing in the ecosystem, not a specific plant type. We discussed this question in the description paper of the ORCHIDEE-PEAT model published by GMD in 2018 (Qiu et al., 2018, GMD: https://www.geosci-model-dev.net/11/497/2018/). Here we cite the discussion in that paper: "*At present, however, ORCHIDEE-PEAT lacks representation of dynamic moss and shrub covers, and we do not know the fractional coverage of different vegetation types at each site in grid-based simulations. Previous studies have shown that there was considerable overlap between the plant traits ranges among different plant functional types, while variations in plant traits within PFTs can be even greater than the difference in means among PFTs (Verheijen et al., 2013; Wright et al., 2005; Laughlin et al., 2010). Therefore, for simplicity, we applied the PFT of C3-grass with a shallower rooting depth to represent the average of vegetation growing in northern peatlands.*
*Only one key photosynthetic parameter—$V_{cmax}$ of this PFT has been tuned to match with observations at each site. This simplification may cause discrepancies between model output and observations. Druel et al. (2017) added non-vascular plants (bryophytes and lichens), boreal grasses, and shrubs into ORC-HL-VEGv1.0. Their work is in parallel with our model and will be incorporated into the model in the future. It will then be possible to verify how many plant functional types are needed by the model to reliably simulate the peatlands at site-level and larger scale.*"

To address this question, we recall the Qiu et al. 2018 description paper on Line117: "……. Vegetations growing in peatlands are represented by one C$_3$ grass plant functional type (PFT) with shallow roots (see dedicated section 2.2.1 of Qiu et al. (2018) for additional discussion on peatland PFT) …"

**2. line 224 - Since Fan et al. 2013 is a model-based product perhaps add in 'simulated' in the description.**

Corrected now in the text on Line224: "……, with areas that have shallow (WT≤20cm) water tables from groundwater modeling of Fan et al. (2013)."

**3. line 265 - Does the peatland HSU immediately shrink to the new potential peatland area fraction? No lag or delay?**

Yes, the peatland HSU immediately shrink to the new potential peatland area fraction, there is no lag or delay. Please see our response to Q8.

**4. line 282 - Why is the old peat treated as mineral soils? That strikes me as strange. The soils would continue to have high C contents for quite a while if drained so treating as a mineral soil seems unreasonable. Please expand on this logic.**

We would like first to note that when simulated peatland area contracts, peat C is still there, not released immediately. But the hydrology of the old peat and the decomposition of C of the old peat is treated as mineral soils. It is noteworthy that draining of peatland may cause decrease of porosity and saturated moisture content. Changes of physical (and chemical) properties of peat soil due to drainage/drought depend on peat type, drainage intensity (Oleszczuk & Truba, 2013; Mustamo, 2017) and duration of drought period. In this study, parameterizations and parameters for old peat and mineral soils are identical, following the study of Stocker et al. (2014). To have a more realistic representation of "old peat soils", the model structure needs to be improved by adding a new sub-grid hydrological soil unit (HSU) which would take hydrological properties of drained peat soils. Substantial original work is needed to change the model structure and to tackle the issue of representation of drained peat soils, thus couldn't be resolved in this study.

We add these sentences on Line282 to acknowledge this issue: "……During the simulation, the contracted area and C are allocated to an 'old peat' pool and are kept track of by the model. It should be noted that drainage (drought) may cause decrease of porosity and saturated moisture content of peat soils (Oleszczuk & Truba, 2013) and, changes in peatland vegetation compositions (Benavides, 2014). But the current model structure doesn't allow us to take these potential changes in peatland into consideration. Therefore, parameterizations of the "old peat" pool is identical to mineral soils, following the study of Stocker et al. (2014). When peatland expansion happens, the peatland will first expand into this 'old peat' area and inherit its stored C (Stocker et al., 2014)."

**5. line 400 - Didn't understand the last sentence there.**

We meant to say that in grid cell G1 and grid cell G3, observed C fraction of peat cores are much larger than median values (obtained from 39 peat cores) we used to calculate empirical amount of C that each model layer can hold in Sect. 2.1.2. Therefore, we can see that in these two gridcells (Fig.3), simulated C concentration along the peat profile are smaller than observations, but peat depth are still overestimated by the model. This happens with grid cell Lake 785 and Lake 396 (Fig.2) and has been described on Line385-394. To clarify, we rephrase the sentence on Line400 as: "……Observed C fraction at grid cell G1 and G3 are much greater than the median value of all peat core samples (Sect. 2.1.2), thus simulated C concentration along the peat profile are smaller than observations, but peat depth are still overestimated by the model. As it is the case with Lake 785 and Lake 396."

**6. line 447 - How many cores were simulated as non-peat out of the total?**
Please see data in the table below: There are in total 1685 and 130 observed peat cores, respectively, in NA and WSL, respectively, from Gorham et al. (2007, 2012) and Kremenetski et al. (2003). Because our study aimed to reproduce development of northern peatlands since Holocene, observed peat cores that are older than 12 ka are removed from the evaluation. Then, 1202 out of 1521 peat cores in NA, and 109 out of 127 peat cores in WSL are captured by the model. In other words, out of 596 gridcells (1° × 1°) that contain observed peat cores in NA, the model simulate peatland in 429 gridcells; and, out of 60 gridcells that contain observed peat cores in WSL, the model simulate peatland in 54 gridcellls.

| | North Amercia (NA) | West Siberian Lowland (WSL) |
|---|---|---|
| Sources of measured peat cores | Gorham et al. (2007, 2012) | Kremenetski et al. (2003) |
| Total number of observed peat cores | 1685 | 130 |
| Number of observed cores that are younger than 12 ka (Holocene) | 1521 | 127 |
| Number of grid cells (1° × 1°) occupied by observed peat cores (cores that are younger than 12 ka) | 596 | 60 |
| Number of grid cells occupied by simulated peat | 429 (Note: there are 1202 observed peat cores in these grid cells) | 54 (Note: there are 109 observed peat cores in these grid cells) |

To note this issue, we add sentences on Line361: "……but contain more samples and cover larger areas. Note that as this study aims to reproduce development of northern peatlands since the Holocene, peat cores that are older than 12 ka are removed from the model evaluation. At last, 1521 out of 1685 observed peat cores in NA, 127 out of 130 observed peat cores in WSL, are used in model evaluation (Sect. 4.2: Peat depth)." And add sentences on Line445: "……dependent on local conditions, i.e. retreat of glaciers, topography, drainage, vegetation succession (Carrara et al., 1991; Madole, 1976). As a large-scale LSM, the model can't capture every single peatland: 429 out of 596 grid cells that contain observed peat cores in NA are captured by the model, while the model simulates peatlands in 54 out of 60 observed grid cells in WSL. Cores that are not captured by the model are removed from further analysis (319 out of 1521 peat cores in NA, 18 out of 127 peat cores in WSL, are removed)."

**7. around line 476 - please specify 'simulated'. It gets a bit confusing that these are all just model quantities.**

Corrected now in the text on Line476: "......From 1901 to 2009, both simulated net primary production (NPP) and simulated heterotrophic respiration (HR) show an increasing trend"

**8. line 626 - This is where I find the technique a bit confusing. 'We notice a large interannual variability in peatland area'. In reality this is unlikely to be possible given that peat soils are slow to develop and slow to leave. The water-logging is the dynamic aspect. This sort of ties into my main comment #1 above. Please tighten up how this is all defined and referred to.**

We agree with the reviewer that peat soils are slow to develop and slow to leave in reality. Although we set no limitation on peatland expanding/shrinking rate in the model parameterization, intra- and inter-annual changes in simulated peatland area were actually constrained by the "inundation persistency" criterion ($Num$, Sect 2.2.2) and the long-term C balance criterion ($C_{lim}$, Sect 2.2.2). Short-term dry/wet climate couldn't cause significant change of peatland area. As shown in the figure below, simulated historical changes in peatland area and C stocks at the Hudson Bay lowlands (HBL) and the West Siberian lowland (WSL) are indeed gradual and small.

[Figure]

Simulated peatland area at the southeastern US, however, showed a large interannual variability. This is because for an area fraction to be diagnosed as peatland at the southeastern US, it needs to be inundated for more than 240 months in the preceding 30 years ($Num$ = 240 months), making simulated peatland area sensitive to short-term variations in climate. The figure below shows the "inundation persistency" parameter ($Num$) for each grid cell, averaged over 1860-2009. The reviewer is right that the large inter-annual variability of peatland area at the southeastern US is related to the water-logging aspect, we remove the confusing sentence from the manuscript.

[Figure]

**9. Fig 1 - Strange figure. I couldn't figure out the green fade, nor understand how it was giving information. So is the above the green the >100% RMSE? Why a fade? Please rethink this one.**

The same question has been raised by Referee1, we follow his suggestion by replacing Fig 1 with a scatterplot which splits temperate/boreal/arctic and bog/fen.

[Figure]

Fig. 1. Measured and simulated peat depth at 60 peatlands sites (Table S1). Shapes of markers indicate peatland types (bogs, fens, others), colors of markers imply climatic zones (temperate, boreal, arctic) of sites' location.

**10. If Fig 6 is plotted as a simple scatterplot, what does it look like? I understand that Fig 7 is a more detailed look but I wonder if a simple scatter plot could be instructive for any bias.**

We enrich Fig 6 by adding scatter plot of measured VS simulated peat depth.

[Figure]

Fig. 6. (a, b) Measured (M) and simulated (S) mean peat depth at the West Siberian lowlands (a) and North America (b), grouped according to the mean age of peat cores. Measured peat cores are from Gorham et al. (2012) and Kremenetski et al. (2003). The horizontal box lines: the upper line - the 75th percentile, the central line - the median (50th percentile), the lower line - the 25th percentile. The dashed lines represent 1.5 times the IQR. The circles are outliers. Number of included grid cells in each age group is indicated by N. (c, d) The scatter plot of measured and simulated peat depth for the West Siberian lowlands (c) and North America (d). For a grid cell that has multiple measured peat cores, the median depth of all measurements is plotted against the simulated depth in the scatter plot.

**11. Fig 10 - please split into 3 separate bars per time period. I couldn't figure this out. What is the light blue? What is the line midway through 8-10 Age bar meaning?**

Fig 10 was indeed misleading. The light blue, and the line through 8-10 Age bar was a result of color overlay. We split the fig into 3 separate bars, as suggested by the referee. Note that we changed the color of the figure.

[Figure]

Fig. 10. (Grey bars) Percentage of observed peatland initiation in 2000-year bins. Peat basal dates of 1516 cores are from MacDonald et al. (2006), peat basal age frequency of each 2000-year bin is divided by the total peat basal age frequency. (White bars) Percentage of simulated peatlands area developed in each 2000-year bin, deglaciation of ice-sheets is not considered (the model was run with 6 times SubC, 2000 years each time). The peatlands area developed in each bin is divided by the simulated modern (the year 2009) peatlands area. (Black bars) Percentage of simulated peatlands area developed in each 2000-years bin, pattern and timing of deglaciation are read from maps in Fig. S5 and Fig. S6.

**12. supplementary line 11 - So does all of the surface runoff from the grid cell get funnelled into the peatland HSU? Why only surface and not subsurface?**

Yes, all surface runoff from the non-peatland HSUs of the grid cell are routed toward the peatland HSU, with the amount of water to be infiltrate into peat soils being calculated through a time-splitting procedure (d'Orgeval, 2006, Diss. Paris; Qiu et al., 2018, GMD). The referee is right that peatlands (fens) can receive both surface and subsurface water. However, the hydrology of the model splits the lateral fluxes into surface runoff and deep drainage. Subsurface runoff are not explicitly represented in the model and therefore not considered as a source of water funneling into the peatland.

**p.s. Apologies for the slow review. There was some confusion between me and the editorial team if I was providing a review.**

---

## Author Response (AR2)

We thank the anonymous referee and Benjamin Stocker, for providing reports
for the revised version of the manuscript. We greatly appreciate the valuable
comments from Benjamin Stocker on both the revised version and an earlier
version of the manuscript.

In the following, please find our point by point response to the comments in the
report.

• Reviewer's comments are in bold
• Modifications done in the revised manuscript are in blue
• All line numbers refer to the revised manuscript version

**The authors carefully addressed or responded to the comments I had raised on the**
**initial submission. The manuscript is now clearer and the work is better motivated**
**in the context of previous work that has been done on global dynamic peatland**
**modelling.**
**Please consider my comments on specific points raised before and the author's**
**response below. I rated "scientific reproducibility" as "fair" due to the argument**
**made below under 'Q1'.**
**Q1: I can see practical reasons that it's often very challenging to separate model**
**parts so that they can be run as a stand-alone. The authors argue that there is a**
**feedback between hydrology and peatland development that is subject to the**
**processes in the non-peatland part of the model. I am suggesting to ignore this and**
**prescribe soil moisture or whatever is required. The point is that the relevant model**
**parts can be run in a demo setup, not actually reproducing the results presented**
**here exactly. Plug and play.**
In the current manuscript, we provided a link to download freely the code and since our
ORCHIDEE model is organized by modules themselves split into subroutines, any scientist
interested is able to run all the subroutines separately. "Prescribe soil moisture" is hiding
the dependence of soil moisture upon non traceable model equations as there is no direct
observation of soil moisture / water table depth. Further, there is a feedback that if peatland
expand and occupy more space, other PFTs will have to be reduced, which is a process
that can only be accounted for by coupling peat with the rest of the equations of a land
surface model. Since our submitted manuscript is more focus on reproducing peat
dynamics including all the different drivers, we are afraid that adding a "plug play" section
might dilute our main message.
**Q2: Would be worth mentioning how permafrost is dealt with also in the manuscript?**
**Appreciate the inclusion of the evaluation of inundation timing.**
We add the below sentence on Line112: "ORCHIDEE-MICT is an updated version of the
ORCHIDEE land surface model with an improved and evaluated representation of high-
latitude processes. Soil water freezing and melting, and subsequent changes in thermal
and hydrological properties, as well as latent heat release and consumption involved in the
freeze - thaw processes are all simulated by this model (Guimberteau et al., 2018). The
model simulates a more rapid thermal signal propagation, and a reduction in soil water infiltration and movement in a frozen soil (Gouttevin et al., 2012). The model calculates the active layer thickness (ALT) from simulated soil temperatures and adjusts root distribution and soil carbon inputs relative to the ALT to represent impacts of permafrost physics on plant water availability and soil carbon profiles. It is worth mentioning that the model resolves one energy budget for all soil tiles in one gridcell, but soil thermal properties of a specific grid cell is defined as the weighted average of mineral soil and pure organic soil in that grid, with C content of the grid cell derived from the soil organic C map from NCSCD (Hugelius et al., 2013) for permafrost regions and from HWSD (FAO et al., 2012) for non-permafrost regions (Guimberteau et al., 2018). This makes it possible to include the impacts of peat carbon on the gridcell soil thermics."

**Q3: Interesting added text on the isostatic rebound effect and the formation of the Hudson Bay Lowland peatland complex. The essential mechanism of the "sponge-feedback" is that, on peatlands, less water is diverted into runoff, raises the water table, and adds to subsequent inundation. Is it spelled out explicitly in the manuscript, that this effect is accounted for by the model?**

We add sentences on Line137 to spell it out: "The large porosity (0.9 $m^3$ $m^{-3}$) and the large saturated water conductivity (2120 mm $d^{-1}$) of the peatland HSU, as well as the addition of an above-surface water reservoir reduce runoff and increase soil water storage and retention (Qiu et al., 2018). Therefore, the occurrence and expansion of peatland increase the grid cell mean water table and enhance inundation."

**Q4: Ok, added text explaining the necessity and effect of the empirical depth scaling addresses my comment.**

**Q5: This is interesting, that the simulation with fixed peatland extent did no yield faster accumulation. But as authors explain, this may be due to the redistribution of runoff water within the gridcell and C transferred from the mineral soil to the peatland fraction. It sounds like this is a model-specific issue then and cannot easily be resolved in this paper. Therefore, I agree with how it's dealt with in the revised manuscript.**

Thank you.

**Q6: I would recommend stating it in the manuscript as clearly as given in the response to the editor that "we can't compare simulated peat C profile against dated peat cores because our model doesn't track age bins explicitly." The statement now given in the manuscript ("more peat core data collected with more rigorous sampling methodologies") does not reflect this limitation of the model evaluation. I would find it helpful if this limitation was spelled out in order to motivate future research in this direction.**

We rephrase the sentences in the manuscript as: "In addition, we can't compare simulated peat C profile against observed profile from dated peat cores because the model doesn't track age bins explicitly."

**Q9: I do not agree with the statement now made in the manuscript: "The model therefore aims to simulate average peat depth and C profile in a grid location rather than capturing peat inception time and age-depth profiles of peat cores." The timing of inception and C accumulation history (yielding the age-depth profile) are essential for simulating the C cycle effect of peatlands. That's what the model will be used for**

| 88 | **and is used for here (Holocene simulations).** |
| 89 | We realized that our statement was indeed misleading we rephrase on line760: "The model |
| 90 | therefore aims to simulate large-scale average peat depth and C profile rather than |
| 91 | capturing local peat inception time and age-depth profiles at the location of specific peat |
| 92 | cores. Tracers like $^{14}$C are not included in the model, making some site to site evaluation |
| 93 | in particular regarding peat inception time and age-depth profiles of peat cores difficult.". |
| 94 | **Q10: The reason I made this point is to motivate a revision of the manuscript text,** |
| 95 | **so that readers will better understand the model.** |
| 96 | We improve the manuscript text on Line193 as: " We first calculate the empirical carbon |
| 97 | content at each model layer ($C_{obs,l}$) according to measured data from 102 peat cores from |
| 98 | 73 sites (Lewis et al., 2012; Loisel et al., 2014; McCarter and Price, 2013; Price et al., 2005; |
| 99 | Tfaily et al., 2014; Turunen et al., 2001; Zaccone et al., 2011). $C_{obs,l}$ is calculated as: |

$$C_{obs,l} = BD_l \times \alpha_{c,l} \times \Delta Z_l \qquad , \qquad\qquad\qquad (3)$$

[revised manuscript text omitted]